# Consistent Noisy Latent Rewards for Trajectory Preference Optimization in Diffusion Models

**Xiaole Xian**[1,2]  **Xilin He**[1,4]  **Wenting Chen**[3]  **Wenshuang Liu**[2]  **Wenqi Mu**[2]
**Yancheng He**[2]  **Liang Li**[5]  **Yi Zhang**[5]  **Xiangyu Yue**[1†]

[1]MMLab, CUHK  [2]Tencent  [3]CityUHK  [4]MBZUAI  [5]Jiutian, China Mobile

## Abstract

Recent advances in diffusion models for visual generation have sparked interest in human preference alignment, similar to developments in Large Language Models. While reward model (RM) based approaches enable trajectory-aware optimization by evaluating intermediate timesteps, they face two critical challenges: **unreliable reward estimation on noisy latents** due to pixel-level models' sensitivity to noise interference, and **single-timestep preference evaluation** across sampling trajectories where single-timestep evaluations can yield inconsistent preference rankings depending on the selected timestep. To address these limitations, we propose a comprehensive framework with targeted solutions for each challenge. To achieve noise compatibility for reliable reward estimation, we introduce the Score-based Latent Reward Model (SLRM), which leverages the complete diffusion model as a preference discriminator with learnable task tokens and a score enhancement mechanism that explicitly preserves noise compatibility by augmenting preference logits with the denoising score function. To ensure consistent preference evaluation across trajectories, we develop Trajectory Advantages Preference Optimization (TAPO), which strategically performs Stochastic Differential Equations sampling and reward evaluation at multiple timesteps to dynamically capture trajectory advantages while identifying preference inconsistencies and preventing erroneous trajectory selection. Extensive experiments on Text-to-Image and Text-to-Video generation tasks demonstrate significant improvements on noisy latent evaluation and alignment performance. The code is available at [TAPO](#).

## 1 Introduction

Inspired by Reinforcement Learning from Human Feedback (RLHF) advancements in Large Language Models (LLMs)(Schulman et al., 2017; Rafailov et al., 2023; Shao et al., 2024) and diffusion models' success in visual generation(Nichol et al., 2021; Rombach et al., 2022), numerous works (Clark et al., 2023; Fan et al., 2023; Black et al., 2023) have emerged for human preference alignment in diffusion models. These methods fall into two families: *offline data* approaches that learn from human-annotated pairs (Wallace et al., 2024), and *reward-model (RM)* approaches that enable online preference optimization by scoring candidates during training (Liu et al., 2025c; Wang et al., 2025; Liang et al., 2025). While offline methods improve final outputs, they only evaluate clean and fully denoised images, ignoring the noisy latents along the sampling trajectory. This limits their ability to support trajectory-aware optimization. In contrast, RM-based methods have gained traction for their ability to evaluate intermediate timesteps and enable such optimization (Liang et al., 2025; Zhang et al., 2025).

Although RM-based methods can provide preference evaluation for intermediate timesteps, they face two primary challenges. The first challenge concerns **unreliable reward estimation on noisy latents**. Most existing methods (Liu et al., 2025d; Xu et al., 2024a) adapt Vision-Language Models (VLMs) as reward models, which could be sensitive to pixel perturbation and lack principled noise

---

[†] Corresponding author.

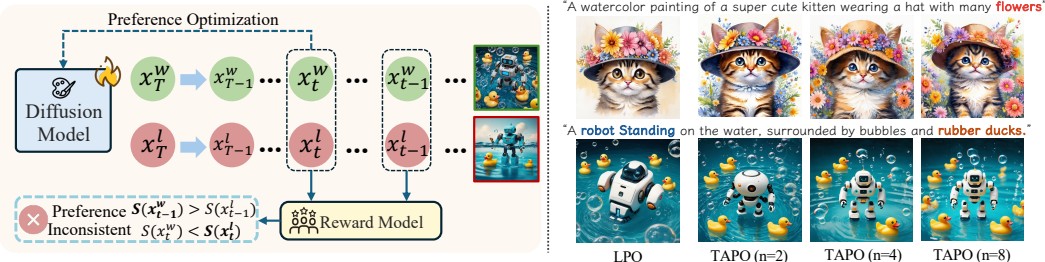

Figure 1: **(a)** Inconsistent reward signals evaluated from different timesteps could lead to incorrect trajectory preference ordering , disrupting the training in existing DPO-style diffusion model optimization. Specific examples are shown in Appendix C.5. **(b) Superior alignment through Trajectory Advantages.** Unlike existing methods (e.g., LPO) that rely on single-step noisy latent evaluation, TAPO leverages multi-steps (n) advantage across the entire sampling trajectory to obtain higher quality training samples, thereby achieving optimal performance.

compatibility. To enhance noise compatibility, recent efforts Liang et al. (2025) inject noise to the visual encoder inputs, with LPO Zhang et al. (2025) further leveraging the diffusion backbone to incorporate stronger noise-aware priors. However, LPO overlooks that the diffusion model's noise compatibility fundamentally stems from learning a score function over the data distribution, which is a fundamentally different objective from reward modeling. Therefore, when these backbones are fine-tuned for preference discrimination, their score-learning properties degrade, resulting in poor performance on noisy latents and unreliable reward predictions.

Another challenge involves **single timestep preference evaluation**. As illustrated in Fig. 1 (a), when comparing noisy latents from two samples, the reward model may prefer the second sample at an intermediate timestep ($S(x_t^W) < S(x_t^l)$) while preferring the first sample at the other timestep ($S(x_{t-1}^W) > S(x_{t-1}^l)$). Current RM-based methods (Liu et al., 2025d; Xu et al., 2024a) typically evaluate preferences at only one specific timestep, failing to consider the full temporal context of the sampling trajectory. Although recent work (Yang et al., 2024b) has explored dense rewards along the trajectory, it redistributes single clean-image preference evaluation to all timesteps. This single-timestep evaluation can yield inconsistent outcomes depending on which timestep is selected, leading to erroneous preference rankings where trajectories with high intermediate rewards but suboptimal final outputs are favored. Thus, considering trajectory-level rewards for preference evaluation is crucial to preventing misleading trajectory selection during training.

To address these challenges, we propose a comprehensive framework for human preference alignment applicable to diffusion-based models, encompassing both reward model training and the online sampling strategy during the alignment phase. First, to **achieve noise compatibility in noisy latents**, we introduce a *Score-based Latent Reward Model (SLRM)*, which leverages the complete diffusion model as a preference discriminator. SLRM introduces learnable task tokens, leveraging self-attention for adaptive, multi-layer aggregation of fine-grained visual and textual features (Peebles & Xie, 2023). Crucially, we incorporate a score enhancement mechanism that explicitly preserves the model's noise compatibility by augmenting the preference logits with the denoising score function. This design ensures that SLRM maintains stable and accurate discriminative evaluations throughout all timesteps. Building on this noise-compatible reward model, we propose *Trajectory Advantages Preference Optimization (TAPO)* to **establish trajectory-level preference evaluation and ensure consistent preference rankings**. TAPO strategically performs Stochastic Differential Equations (SDE) sampling for stochastic exploration (Song et al., 2020) and reward evaluation at selected multi-timesteps, retaining the best and worst samples at each evaluation step based on the reward. This dynamically captures trajectory advantages while avoiding the computational overhead of exhaustive evaluation. Moreover, in Fig. 1 (b), it efficiently identifies win-lose sampling data with pronounced quality differences, yielding high-quality training data for more accurate preference alignment. Extensive experiments on Text-to-Image (T2I) and Text-to-Video (T2V) generation show our method substantially improves visual generation quality. SLRM achieves significant accuracy improvements across all sampling timesteps. During alignment, TAPO demonstrates substantial generation quality improvements when applied to Stable Diffusion-3.5 (T2I) (Esser et al., 2024) and Wan-2.1 (T2V) models (Wan et al., 2025).

Our contributions are as follows:

- We introduce SLRM, a score-based noisy latent reward model that leverages the diffusion model's inherent score function to maintain noise compatibility throughout all timesteps, addressing the critical limitation of existing pixel-level reward models in evaluating intermediate noisy latents.

- We propose the TAPO that strategically performs SDE sampling and reward evaluation at multiple timesteps, dynamically capturing the trajectory advantages to generate high-quality training data with pronounced preference differences.

- Through extensive experiments across two generation tasks (T2I and T2V), we demonstrate the significant improvements in generation quality, establishing the broad applicability of our approach to diverse diffusion-based models.

## 2 RELATED WORK

**Human Preference Alignment for Diffusion Models.** Motivated by RLHF's success in LLMs (Schulman et al., 2017; Achiam et al., 2023; Shao et al., 2024; Chen et al., 2024), extensive research has explored preference alignment for diffusion models (Rombach et al., 2022; Nichol et al., 2021; Ramesh et al., 2021; Saharia et al., 2022). These approaches fall into two categories: offline data methods and reward-model (RM) based methods. Early RM-based approaches used PPO-based policy gradients (Fan et al., 2023; Black et al., 2023), formulating denoising as a Markov decision process, while reward-driven fine-tuning methods (Li et al., 2025a; Lee et al., 2025; Xu et al., 2023; Wu et al., 2023; Ma et al., 2025) directly optimize diffusion models to maximize reward signals. However, these methods suffer from reward hacking and expensive computational costs. Following Diffusion-DPO (Wallace et al., 2024), which adapted DPO (Rafailov et al., 2023) to diffusion models, subsequent works (Liu et al., 2025d;c; Wang et al., 2024a; Zhang et al., 2024a; Lu et al., 2025; Wu et al., 2025) have advanced preference alignment through implicit reward modeling. While offline methods (Wallace et al., 2024) improve outputs by learning from human-annotated pairs, they evaluate only clean images and cannot assess noisy latents along sampling trajectories. Consequently, RM-based methods are increasingly adopted for trajectory-aware optimization (Liang et al., 2025; Zhang et al., 2025), enabling online preference optimization by scoring intermediate timesteps. However, these methods face challenges with unreliable rewards on noisy latents and inconsistent preference evaluation across trajectories.

**Reward Model for Preference Optimization.** Reward models provide crucial feedback signals for preference-based optimization in generative model alignment. Early approaches leverage pre-trained VLMs like CLIP (Radford et al., 2021) and BLIP (Li et al., 2022) for zero-shot evaluation, or employ fine-tuned models (Xu et al., 2023; Wu et al., 2023; Ma et al., 2025; Kirstain et al., 2023; Zhang et al., 2024b; Li et al., 2025a) for aesthetic and preference assessment. Recently, LLM-based reward models (Liu et al., 2025c; Wang et al., 2024b; Xu et al., 2024b) have emerged, leveraging MLLMs' contextual understanding for alignment evaluation. However, pixel-level reward models face limitations when evaluating intermediate noisy latents during denoising, as most existing methods (Liu et al., 2025d; Xu et al., 2024a) could be sensitive to pixel variations and unable to handle noise effectively. Recent attempts (Liang et al., 2025; Zhang et al., 2025) train latent reward models on simulated noisy inputs but suffer from noise compatibility degradation due to insufficient understanding of the diffusion model's score function (Song et al., 2020). These limitations motivate our SLRM proposal, which directly incorporates the denoising score function to maintain noise compatibility. Furthermore, current methods determine the advantages of the whole sampling trajectory based on a single point-wise timestep, neglecting the temporal context from the global trajectory-level perspective. Thereby, we further propose TAPO to consider reward signals from all timesteps when determining the win-lose trajectory.

## 3 METHODOLOGY

In Fig. 2, we present a comprehensive two-stage framework for human preference alignment. First, we propose the Score-based Latent Reward Model (SLRM) in Sec. 3.2 for stable preference discrimination across all timesteps. Subsequently, building upon this noise-compatible reward model, we

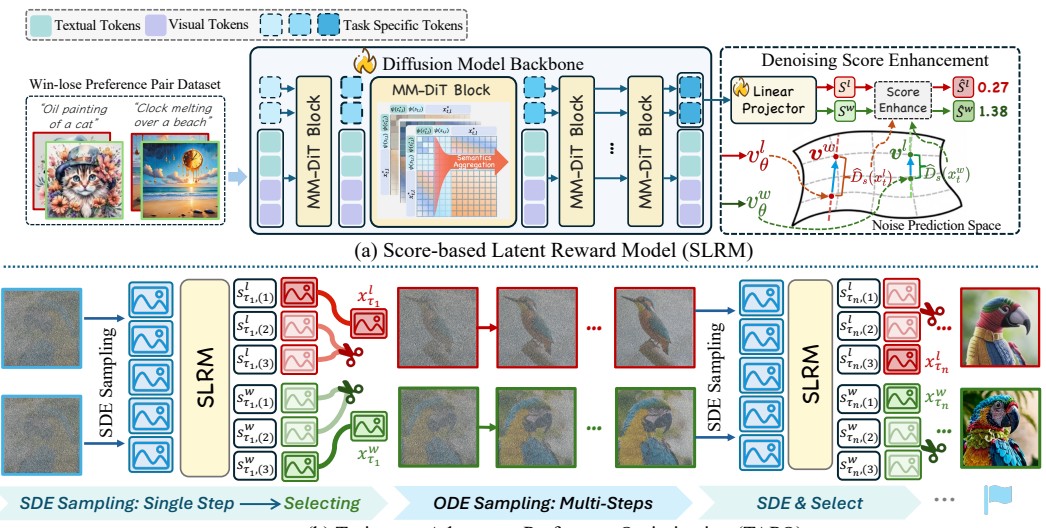

Figure 2: (a) Training Pipeline of the Score-based Latent Reward Model (SLRM). (b) TAPO sampling process. At specific timesteps, win and lose latents adapt SDE sampling to get the latents group respectively.

introduce Trajectory Advantages Preference Optimization (TAPO) in Sec. 3.3. It leverages SLRM to evaluate samples along sampling trajectories, efficiently identifying win-lose pairs with pronounced preference differences.

## 3.1 PRELIMINARY

**Flow Matching.** Suppose that $x_0 \sim X_0$ is a data sample from target distribution and $x_1 \sim X_1$ denotes the source distribution. Recent advanced diffusion models adopt the flow matching (Lipman et al., 2022) to generate $x_0$ starting from $x_1$. Specifically, the flow matching framework defines a continuous-time normalizing flow through an ordinary differential equation (ODE):

$$\mathrm{d}\boldsymbol{x}_t = \boldsymbol{v}_t \mathrm{d}t \tag{1}$$

where the linear conditional flow defines the $\boldsymbol{x}_t = (1-t)\boldsymbol{x}_0 + t\boldsymbol{x}_1$. The core of these methods is to train a neural network $v_{t,\theta}$ to satisfy the velocity field by minimizing the Flow Matching objective:

$$\mathcal{L}_{FM} = \mathbb{E}_{t \in [0,1], \boldsymbol{x}_t \sim p_t} \|\boldsymbol{v}_t(\boldsymbol{x}_t) - \boldsymbol{v}_\theta(\boldsymbol{x}_t)\|^2. \tag{2}$$

where the velocity field is given by $v_t(x_t) = \boldsymbol{x}_1 - \boldsymbol{x}_0$.

**Preference Optimization for Diffusion Models.** Diffusion-DPO (Wallace et al., 2024) extends DPO (Rafailov et al., 2023) to diffusion models by propagating preference orders from clean images $(\boldsymbol{x}_0^w, \boldsymbol{x}_0^l)$ to latents in intermediate denoising steps $(\boldsymbol{x}_t^w, \boldsymbol{x}_t^l)$. However, the preference orders may be inconsistent along the all the timesteps, which has motivated subsequent work to directly evaluate the preference orders of latents in intermediate steps $(\boldsymbol{x}_t^w, \boldsymbol{x}_t^l)$. Accordingly, the optimization objective of Diffusion-DPO is reformulated as a step-by-step preference optimization (SPO):

$$\mathcal{L}_{SPO} = -\mathbb{E}_{\boldsymbol{x}_t^w, \boldsymbol{x}_t^l \sim p_\theta(\boldsymbol{x}_t | \boldsymbol{x}_{t+1}, c)} \left[ \log \sigma \left( \beta \log \frac{p_\theta(\boldsymbol{x}_t^w | \boldsymbol{x}_{t+1}, c)}{p_{ref}(\boldsymbol{x}_t^w | \boldsymbol{x}_{t+1}, c)} - \beta \log \frac{p_\theta(\boldsymbol{x}_t^l | \boldsymbol{x}_{t+1}, c)}{p_{ref}(\boldsymbol{x}_t^l | \boldsymbol{x}_{t+1}, c)} \right) \right]. \tag{3}$$

where $c$ is the input condition, $\beta$ is a regularization hyperparameter, and $p_\theta$ and $p_{ref}$ denote the optimized and reference model, respectively.

## 3.2 SCORE-BASED LATENT REWARD MODEL

In this section, we provide a detailed presentation of the overall Score-based Latent Reward Model (SLRM), including the architecture, loss and the training process.

**Architecture Design.** To inherit the diffusion model's capability of processing noisy inputs, our SLRM directly initialized the complete pre-trained diffusion model as the backbone, as shown in Fig. 3. Previous approaches that compute scores using separate visual and text encoders:

$$S(x^*, c) = \left\langle \frac{E_{\text{vis}}(x^*)}{\|E_{\text{vis}}(x^*)\|_2}, \frac{E_{\text{txt}}(c)}{\|E_{\text{txt}}(c)\|_2} \right\rangle, \quad * \in \{w, l\} \tag{4}$$

where the $E_{vis}$ and $E_{txt}$ denote visual and text encoders respectively, and $w$, $l$ represent winning or losing inputs. However, computing preference scores based on visual-textual similarity primarily measures text-image alignment rather than comprehensive quality aspects like fine-grained details and aesthetics.

Thus, we introduce specific task tokens (Xu et al., 2024b) to participate in the model's self-attention process to capture comprehensive quality aspects beyond simple text-image alignment.

Specifically, we randomly initialize these tokens as new embeddings $s \in \mathbb{R}^{n_s \times n_p}$ of the text encoder. Subsequently, these task tokens are concatenated with text and visual tokens to pass through the diffusion model to participate in its self-attention, and the attention features can be modified as follows:

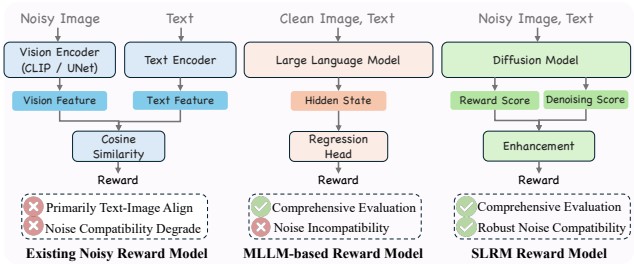

Figure 3: **Comparison of Different Reward Model Architectures.** SLRM maintains robust noise compatibility while enabling comprehensive evaluation through denoising score enhancement.

$$
\begin{aligned}
Q &= P_Q^I(x_{t,l}^*) \odot P_Q^T(\psi(c_{t,l})) \odot P_Q^S(\psi(s_{t,l})), \\
K &= P_K^I(x_{t,l}^*) \odot P_K^T(\psi(c_{t,l})) \odot P_K^S(\psi(s_{t,l})), \\
V &= P_V^I(x_{t,l}^*) \odot P_V^T(\psi(c_{t,l})) \odot P_V^S(\psi(s_{t,l})),
\end{aligned} \tag{5}
$$

where the $P_Q^I, P_K^I, P_V^I$ and $P_Q^T, P_K^T, P_V^T$ are the pre-trained linear projections for image and text embeddings, $P_Q^S, P_K^S, P_V^S$ are the score linear projections. After processing through $L$ layers of the MM-DiT blocks (Esser et al., 2024) in diffusion model, we obtain $s_{t,L}$, which is then mapped to the vanilla reward score via a linear layer:

Through the self-attention mechanism in these DiT blocks, these task tokens can adaptively select and aggregate and fuse the multi-level semantic representations in visual and textual features.

**Denoising Score Enhancement.**

Although our model initially inherits timestep-aware capability from the pre-trained diffusion model, we observe a critical issue: the model's ability to discriminate preferences on noisy latents actually degrades as training, as illustrated in Fig. 4.

This degradation occurs because diffusion models' noise compatibility stems from their original training objective of learning score functions (Song et al., 2020) across different noise levels. However, when we adapt these models for preference discrimination, the training objective fundamentally shifts away from score function learning to preference ranking. Essentially, this results in a naive SLRM, similar to previous works (Liang et al., 2025; Zhang et al., 2025; Dhariwal & Nichol, 2021) that attempted to achieve noise compatibility by simply adding noise to their inputs.

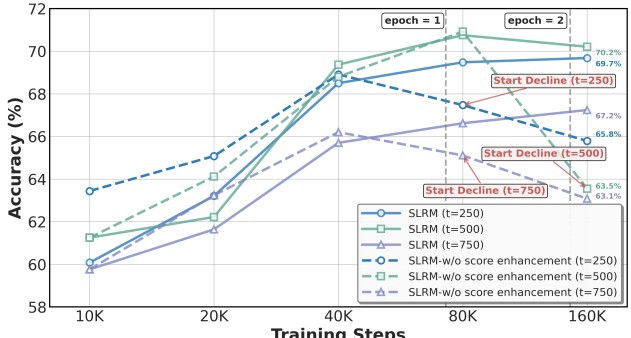

Figure 4: **Impact of Score Enhancement on SLRM.** It compares SLRM with and without score enhancement across different timesteps.

To resolve the gradual degradation during preference learning in existing methods, we design the denoising score enhancement mechanism that maintains the model's noise compatibility by incorporating denoising score matching into preference evaluation. Specifically, we first compute the denoising score matching distance $D_s(x, y, s)$ of the diffusion model. For a DiT-based model with flow matching, this is expressed as:

$$D_s(x, y, s) = \mathbb{E}_{t \sim \mathcal{U}(0,1), \boldsymbol{x}_1 \sim \mathcal{N}(0,\mathrm{I})}[\|v_\theta(\boldsymbol{x}_t, t, c, s) - (\boldsymbol{x}_1 - \boldsymbol{x}_0)\|^2] \tag{6}$$

where $s$ is the score embeddings. For computational efficiency with individual samples, we follow (Lee et al., 2023) and use the estimation for this expectation and modify the distance as:

$$\widehat{D}_s(x, y, s) = e^{-\rho \cdot \|v_\theta(\boldsymbol{x}_t, t, c, s) - (\boldsymbol{x}_1 - \boldsymbol{x}_0)\|^2} \tag{7}$$

where $\rho$ is the scale logit to ensure the score distance is scale-compatible with $S(x_t, c)$. Finally, we use this distance to augment the score logit $S(x_t, c)$:

$$\hat{S}(x_t, c) = S(x_t, c) \cdot \widehat{D}_s(x_t, c, s) \tag{8}$$

The reward score $S(x_t, c)$ corrected by the denoising score must not only consider the aggregated semantic information across variant blocks, but also adapt based on the denoising viability of the latent at its current noise level.

**Training of SLRM.** Building on how the denoising score enhanced the predicted reward score as described above, we now detail the training loss of SLRM. Following prior works (Liu et al., 2025c; Yang et al., 2024a) that train a Bradley-Terry (BT) style reward model, we adopt a contrastive learning approach for optimization. Given a preference dataset $\mathcal{P} = \{(x_i^w, x_i^l, c_i)\}_{i=1}^N$, where $x^w$ and $x^l$ are win-lose pair images corresponding to the same prompt $c$. We randomly sample random timesteps $t \in \{1, \ldots, T\}$, where $T$ is the number of timesteps. The paired images are transferred into noisy latents $x_t^w$ and $x_t^l$ through the scheduler. These noisy latents are then fed into Eq. 9 to obtain the predicted scores $\hat{S}(x_t^w, c)$ and $\hat{S}(x_t^l, c)$ respectively, and the training loss of our reward model is formulated as:

$$\mathcal{L}_{SLRM} = -\mathbb{E}_{t \sim \mathcal{U}(0,T), (x^w, x^l, c) \in \mathcal{P}} \log \frac{\hat{S}(x_t^w, c)^\eta}{\hat{S}(x_t^w, c)^\eta + \hat{S}(x_t^l, y)^\eta}, \tag{9}$$

## 3.3 Trajectory Advantage Sampling for Preference Optimization

During the alignment phase, existing optimization methods typically sample two trajectories to form a win-lose pair $(\boldsymbol{x}^w, \boldsymbol{x}^l)$ and determine preference order based on the reward on single intermediate timesteps. Since previous work (Liu et al., 2025a; 2024) indicates that diffusion models focus on different dimensions at different timesteps (e.g., layout and composition in early stages, content coherence in middle stages, and visual details in late stages), our SLRM is designed to effectively evaluate intermediate latents along sampling paths, thereby capturing comprehensive trajectory advantages beyond single-step signals. However, fully leveraging these advantages through exhaustive evaluation presents a critical trade-off with computational efficiency.

To balance this, we propose Trajectory Advantages Preference Optimization (TAPO), a sampling strategy that efficiently identifies and amplifies trajectory advantages by strategically performing multi-step evaluations and progressively pruning less preferred samples. Specifically, within the total $T$ sampling timesteps, we first designate $n$ sampling steps where SDE sampling is performed to introduce randomness for stochastic exploration. They are uniformly distributed across the sampling trajectory, with the corresponding timestep set $W_T$ defined as:

$$W_T = \{\tau_1, \tau_2, ..., \tau_n\} \subset \{1, 2, ..., T\} \quad and \quad \tau_i = t_{init} + \lfloor \frac{(i-1) \cdot T}{n} \rfloor, \quad i \in \{1, 2, ..., n\}. \tag{10}$$

where $t_{init}$ denotes the initial evaluation timestep. While the remaining steps use strategy of ordinary differential equation (ODE) to provide deterministic path for efficient sampling. The sampling process can be formulated as:

$$\boldsymbol{x}_t = \begin{cases} \boldsymbol{x}_{t+1} - \left[\boldsymbol{v}_\theta(\boldsymbol{x}_{t+1}, t) + \frac{\sigma_t^2}{2t}\left(\boldsymbol{x}_{t+1} + (1-t)\boldsymbol{v}_\theta(\boldsymbol{x}_{t+1}, t)\right)\right]\phi(t) + \sigma_t\sqrt{\phi(t)}\epsilon, & \text{if } t \in W_T \\ \boldsymbol{x}_t - \boldsymbol{v}_\theta(\boldsymbol{x}_t, t)\phi(t), & \text{otherwise} \end{cases} \tag{11}$$

where $\phi(t)$ denotes the timestep interval determined by the scheduler, $\epsilon \sim \mathcal{N}(0, \boldsymbol{I})$ injects stochasticity. $\sigma_t$ is the parameter controls the level of stochasticity during generation. At each SDE sampling step, we obtain their respective sets of winning latent candidates $\mathbb{X}_t^w = \{x_{t,(i)}^w\}_{i=1}^P$ and losing latent candidates $\mathbb{X}_t^l = \{x_{t,(i)}^l\}_{i=1}^P$. We evaluate them using SLRM and further retaining only the highest and lowest scoring samples:

$$
\begin{aligned}
s_{t,(i)}^* &= \hat{S}(x_{t,(i)}^*, y, t), \quad i \in \{1, ..., P\}, \quad * \in \{w, l\}, \\
x_t^w &= \operatorname{argmax}_{x \in \mathbb{X}_t^w} s_{t,(i)}^w, \quad x_t^l = \operatorname{argmin}_{x \in \mathbb{X}_t^l} s_{t,(i)}^l
\end{aligned}
\tag{12}
$$

After selecting the samples produced by the SDE, the two samples are selected for the subsequent ODE sampling phase. This process repeats iteratively, ultimately yielding the optimal and worst samples at the end of the sampling trajectories.

Notably, the $n$ SDE sampling steps are uniformly distributed across the trajectory, covering diverse noise levels from early to late denoising phases. This enables progressive identification of distinct advantages from coarse to fine-grained across generation phases. Based on this insight, we dynamically capture trajectory advantages and identify win-lose pairs with pronounced quality differences while avoiding the computational burden of exhaustive per-step evaluation. This approach provides stronger training pair samples for preference optimization.

Following the SPO (Liang et al., 2025) framework, we then optimize the model using the win-lose pairs have been collected:

$$
\mathcal{L}_{TAPO} = -\mathbb{E}_{t \in W_T, x_T \sim \mathcal{N}(\mathbf{0}, \mathbf{I}), \boldsymbol{x}_{t-1}^w, \boldsymbol{x}_{t-1}^l \sim p_\theta(\boldsymbol{x}_{t-1}|c, t, \boldsymbol{x}_t)}
$$
$$
\left[ \log \sigma \left( \beta \log \frac{p_\theta\left(\boldsymbol{x}_{t-1}^w | c, t, \boldsymbol{x}_t\right)}{p_{\text{ref}}\left(\boldsymbol{x}_{t-1}^w | c, t, \boldsymbol{x}_t\right)} - \beta \log \frac{p_\theta\left(\boldsymbol{x}_{t-1}^l | c, t, \boldsymbol{x}_t\right)}{p_{\text{ref}}\left(\boldsymbol{x}_{t-1}^l | c, t, \boldsymbol{x}_t\right)} \right) \right].
\tag{13}
$$

We summarize the training procedure of TAPO in Algorithm. 1, can be found in Appendix. B.4.

## 4 EXPERIMENTS

### 4.1 EXPERIMENTAL SETUP

**Implementation Details.** For T2I, we employ SD3.5-medium (Esser et al., 2024) as our base model for both SLRM and TAPO, while we utilize Wan2.1-1.3B (Wan et al., 2025) as our base model for T2V. More training details and comparison methods setting can be seen in Appendix B.1. We primarily validate the effectiveness of our preference alignment method on T2I, while also conducting experiments on T2V to demonstrate the effectiveness of our approach.

**Datasets.** For **T2I**, SLRM is trained on Pick-a-Pic v1 dataset (Kirstain et al., 2023) (580k preference pairs) and evaluated on its validation/test sets (28k) for win-lose discrimination accuracy. For fairness, TAPO uses 4k prompts in SPO for online sampling with 20 timesteps. **For T2V**, due to the lack of high-quality video preference datasets, we collected a preference dataset (10k pairs) for SLRM training and evaluate on GenAI-Bench (1.9k samples). Dataset details are in Appendix B.2.

**Evaluation Metrics.** For T2I, we evaluate TAPO on (1) text-image alignment using CLIP Score(Radford et al., 2021) and GenEval(Ghosh et al., 2023), and (2) general preference using PickScore(Kirstain et al., 2023), HPSv2.1(Wu et al., 2023), HPSv3(Ma et al., 2025), and MPS (Zhang et al., 2024b). All metrics are evaluated on Pick-a-Pic v1 validation set. Result of additional benchmarks of T2I and T2V can be seen in Appendix C.2.

### 4.2 QUANTITATIVE EVALUATION

**Comparison with SOTA Alignment Methods.** Our method demonstrates substantial performance improvements across multiple evaluation dimensions. As shown in Tab. 1, TAPO achieves state-of-the-art performance across most evaluation metrics. Our method attains the best results on both general preference and text-image alignment metrics. Particularly on HPSv3, the latest preference alignment metric, TAPO outperforms the preference alignment method LPO and the latest FLUX.1 Dev by 0.79 and 0.70, respectively. It demonstrates that our method achieves the best overall performance in visual quality and aesthetic preference. The user study is presented in the Appendix C.1.

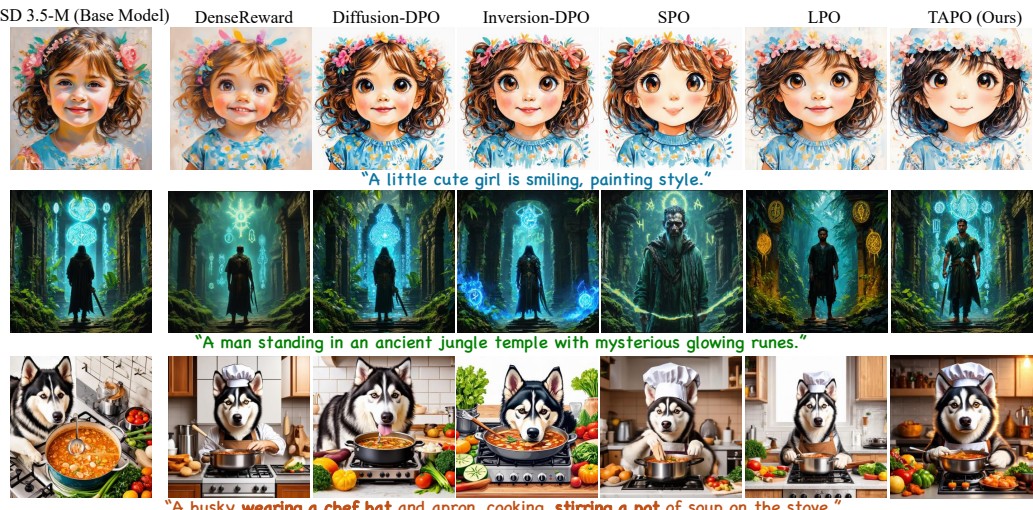

Figure 5: **Comparison with SOTA (T2I).** Qualitative comparison among various preference optimization methods based on SD 3.5-M (Esser et al., 2024). TAPO achieves superior overall generation quality, outperforming baseline methods (DiffusionDPO, InversionDPO, SPO, LPO) in text alignment, visual quality, and aesthetic preference.

Table 1: **Quantitative comparison results on Pick-a-Pic validation unique set.**

| Method | | General Preference | | | | T2I Alignment | |
|---|---|---|---|---|---|---|---|
| | | HPSv2.1 (↑) | HPSv3 (↑) | PickScore (↑) | MPS (↑) | CLIP-Score (↑) | GenEval (↑) |
| **Base Model** | SD-XL (Podell et al., 2023) | 26.05 | 7.52 | 21.94 | 0.89 | 24.73 | 52.29 |
| | SD3.5-M (Esser et al., 2024) | 27.15 | 8.29 | 22.15 | 1.00 | 25.18 | 55.34 |
| | FLUX.1 Dev (Labs, 2024) | 30.08 | 9.19 | 22.72 | 3.29 | 26.08 | 58.20 |
| | Show-o (Xie et al., 2024) | 25.17 | 8.05 | 20.96 | 0.83 | 24.32 | 46.92 |
| **Alignment Model** | Diffusion-DPO (Wallace et al., 2024) | 28.23 | 7.59 | 22.64 | 1.93 | 25.71 | 54.93 |
| | Inversion-DPO (Li et al., 2025b) | 30.83 | 7.91 | 22.91 | 2.08 | 25.76 | 52.34 |
| | DenseReward (Yang et al., 2024b) | 29.99 | 8.05 | 22.83 | 2.51 | 26.18 | 55.27 |
| | SPO (Liang et al., 2025) | 31.52 | 8.74 | 22.70 | 2.24 | 24.72 | 52.75 |
| | LPO (Zhang et al., 2025) | 31.89 | 9.10 | 22.86 | **3.12** | 26.15 | 59.85 |
| | **TAPO (Ours)** | **32.01** | **9.89** | **23.03** | 3.07 | **27.07** | **68.93** |

**Comparison with SOTA Reward Model.** To verify the noise compatibility of our SLRM reward model, we compare it with existing reward models at different timesteps. As shown in Tab. 2, the results demonstrate that our method achieves significantly higher accuracy under noisy inputs and perform better on evaluate the noisy latents. Our SLRM maintains high accuracy of 62.09% and 65.50% under these conditions. Compared to the diffusion-based LRM-3.5, SLRM achieves superior performance across all timesteps, validating the effectiveness of our proposed score-enhanced learning strategy. Although HPSv3 and PickScore achieve higher accuracy of 72.80% and 71.93% respectively on clean images, SLRM's discrimination under noise input far exceeds theirs.

**Noisy Compatibility of SLRM.** To validate SLRM's discriminative capability across the denoising trajectory, we compare with existing noise-compatible methods across comprehensive timesteps. As shown in Tab. 4, our method achieves the best performance with 70.21% accuracy at $t = 501$, demonstrating SLRM's ability to assess noisy latents. This robust cross-timestep performance is crucial for effective reward scoring in subsequent optimization.

## 4.3 QUALITATIVE EVALUATION

**T2I.** We qualitatively compare our method with the SOTA DPO-style approaches, including DiffusionDPO (Wallace et al., 2024), Inversion-DPO (Li et al., 2025b), SPO (Liang et al., 2025) and LPO (Zhang et al., 2025). While DiffusionDPO and InversionDPO improve detail and color quality, they exhibit insufficient text alignment (3-$rd$ row). SPO and LPO enhance text alignment, but SPO over-emphasizes subjects (1-$st$ row) with degraded visual quality (2-$nd$ row), while LPO shows insufficiency in aesthetic quality. Our method successfully balances text alignment, visual quality, and aesthetic preference, generating superior overall quality across diverse scenarios.

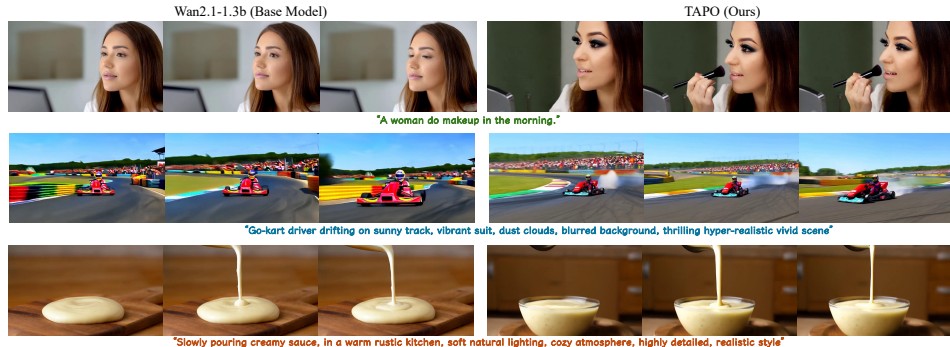

Figure 6: **Results of TAPO in Text-to-Video.**

Table 2: **Comparison with Existing Reward Models.** Accuracy comparison of VLM-based and diffusion-based reward models at different timesteps on Pick-a-Pic validation and test sets.

| | Model | $t \in [501, 1000]$ | $t \in [1, 500]$ | $t = 0$ (Clean Image) |
|---|---|---|---|---|
| VLM-Based | Aesthetic | 47.25 | 45.83 | 54.03 |
| | CLIP Score | 46.91 | 44.37 | 61.84 |
| | VQAScore | 48.12 | 46.55 | 59.16 |
| | ImageReward | 49.68 | 47.92 | 62.66 |
| | HPSv2.1 | 49.31 | 52.04 | 65.58 |
| | HPSv3 | 44.26 | 47.26 | **72.80** |
| | PickScore | 45.26 | 48.21 | 71.93 |
| Diffusion-Based | LRM-3.5 | 59.56 | 64.12 | 66.41 |
| | SLRM(Ours) | **66.35** | **66.59** | 67.08 |

Table 3: **Ablation Study.** Ablation study of SLRM and TAPO. "$n$" indicates the number of SDE sampling. "w/o Task Tokens": use image-text similarity score. "$(\cdot)$ s/iter" denotes time for sampling a pair.

| Strategy | SLRM | TAPO | | |
|---|---|---|---|---|
| | Val-Test Accuracy | MPS | GenEval | HPSv2.1 |
| SLRM (w/o Score Enhanced) | 62.49 | 1.92 | 54.83 | 26.49 |
| SLRM (w/o Task Tokens) | 65.33 | 2.08 | 55.27 | 28.37 |
| SLRM | 67.52 | 3.07 | 64.93 | 30.71 |
| TAPO (n = 2, 3.50s/iter) | - | 2.27 | 52.39 | 27.39 |
| TAPO (n = 4, 3.91s/iter) | - | 2.93 | 56.20 | 28.31 |
| TAPO (n = 8, 4.52s/iter) | - | **3.07** | 68.93 | **32.01** |
| TAPO (n = 16, 6.08s/iter) | - | 2.09 | **69.18** | 31.29 |

Table 4: **Preference Prediction Accuracy across Timesteps.** Results of SLRM's robust performance across detailed denoising timesteps compared to existing methods on Pick-a-Pic.

| Method | Variant Timestep | | | | | | | | | |
|---|---|---|---|---|---|---|---|---|---|---|
| | $t = 1$ | $t = 101$ | $t = 201$ | $t = 301$ | $t = 401$ | $t = 501$ | $t = 601$ | $t = 701$ | $t = 801$ | $t = 901$ |
| SPM | 63.75 | 62.41 | 62.97 | 62.58 | 61.74 | 61.50 | 60.82 | 58.92 | 56.21 | 53.46 |
| LRM-3.5 | 65.42 | 63.78 | 64.25 | 64.03 | 63.12 | 62.89 | 62.15 | 60.28 | 57.64 | 54.83 |
| SLRM (w/o Score Enhanced) | 64.27 | **66.15** | 66.37 | 63.58 | 64.94 | 64.07 | 62.47 | 60.17 | 57.64 | 55.25 |
| **SLRM** | **65.81** | 63.88 | **67.88** | **66.15** | **69.68** | **70.21** | **69.55** | **66.65** | **67.24** | **58.12** |

**T2V.** We qualitatively compare our method with the base model Wan2.1-1.3b (Wan et al., 2025) across diverse video generation scenarios. The base model demonstrates basic generation capabilities but exhibits limitations in temporal coherence, text alignment and aesthetic preference. In contrast, our TAPO consistently generates the videos that are more natural temporal dynamics, enhanced visual details, and superior text-video alignment.

## 4.4 ABLATION STUDY

**Effectiveness of SLRM.** We conducted an ablation study to evaluate the effectiveness of SLRM's different components in Table 2. For SLRM, removing the score enhancement mechanism results in a significant accuracy drop from 67.52% to 62.49% (5.03% decrease), demonstrating its critical role in maintaining noise compatibility. This validates that both components are essential for effective preference discrimination.

**Influence of Evaluation Numbers in TAPO.** We analysed the influence of the number of SDE sampling and reward evaluation steps $n$ (Eq. 10) in TAPO. In Table 3, larger steps allow the model to explore more possibilities, increasing the quality of training sample pairs. Notably, when $n = 8$, GenEval shows a significant improvement, reaching 64.93. When we attempt to sample with more steps, there is no significant improvement in general preference, while sampling time increases substantially ($4.52s/iter \rightarrow 6.08s/iter$). Thus, we choose $n = 8$ as the setting of our main result. More ablation study between the candidate size $P$ and evaluation number $T$ is presented in Appendix C.4.

## 5 Conclusion

We address two critical challenges in diffusion model preference alignment: unreliable reward estimation on noisy latents and inconsistent preference evaluation across sampling trajectories. Our solution introduces SLRM, a score-based reward model that maintains noise compatibility through denoising score enhancement, and TAPO, a trajectory-aware optimization strategy that captures multi-timestep advantages for effective preference learning. Extensive experiments on T2I and T2V tasks demonstrate significant improvements, with SLRM achieving superior performance in noisy latent evaluation and TAPO attaining state-of-the-art results on HPSv3 and GenEval.

## 6 Ethics Statement

This work adheres to the ICLR Code of Ethics. In this study, no animal experimentation was involved. All datasets used, including Pick-a-pic and our collecting video preference dataset, were sourced in compliance with relevant usage guidelines, ensuring no violation of privacy. We have taken care to avoid any biases or discriminatory outcomes in our research process. No personally identifiable information was used, and no experiments were conducted that could raise privacy or security concerns. We are committed to maintaining transparency and integrity throughout the research process.

## 7 Reproducibility Statement

To ensure reproducibility, we have made the following efforts: (1) We will release our code and the collecting dataset. Additionally, the dataset Pick-a-pic are publicly available, ensuring consistent and reproducible evaluation results. (2) We provide experiments setup in Sec. 4 and the more details about training process are presented in Appendix. B.1 including training steps, model configurations, and hardware details. (3) We elaborate on our evaluation protocol in detail in Sec. 4. We believe these measures will enable other researchers to reproduce our work and further advance the field.

## 8 Acknowledgment

This work is partially supported by the National Natural Science Foundation of China (No. 62306261), HK RGC-Early Career Scheme (No. 24211525), ITSP Platform Project (No. ITS/600/24FP), Funding from China Mobile, and the SHIAE Grant (No. 8115074). This study was supported in part by the Centre for Perceptual and Interactive Intelligence, a CUHK-led InnoCentre under the InnoHK initiative of the Innovation and Technology Commission of the Hong Kong Special Administrative Region Government. This work is also partially supported by Hong Kong RGC Strategic Topics Grant (No. STG1/E-403/24-N), and CUHK-CUHK(SZ)-GDST Joint Collaboration Fund (No. YSP26-4760949).

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

# Appendix of "Score-based Enhanced Latent Reward Model for Diffusion-based Visual Generation"

- Section B provides detailed technical details of our methods, including training parameters, construction of the video preference dataset, and derivation of key SDE sampling.
- Section C provides extended experimental results of SLRM and TAPO, including additional benchmark evaluations of both methods on T2I and T2V tasks, as well as ablation studies on hyperparameters during TAPO training.
- Section D provides more visual results of our TAPO.

## A   THE USE OF LARGE LANGUAGE MODELS (LLMs)

Large Language Models (LLMs) were used to aid or polish the writing of this manuscript. Specifically, we used Claude-4-Sonnet solely for language polishing and grammatical refinement of the written text. All research contributions, including the main ideas, technical approaches, experimental work, and scientific insights presented in this paper, are entirely the work of the human authors. The LLM usage is limited to improving the clarity and readability of the already-written content without altering the substance or meaning of our work.

# B  TECHNICAL DETAILS

## B.1  TRAINING DETAILS

To ensure a fair comparison, we retrained the diffusion-based reward model LRM based on SD3.5-medium which was originally based on SDXL-base. Similarly, SPO and LPO were also retrained on SD3.5-medium. All the hyperparameters of our training are shown in the Tab. 5. All the experiments are conducted on 8 NVIDIA A100 GPUs. The optimizer in SLRM and TAPO are both AdamW with default parameters: beta1=0.9, beta2=0.999, weight decay=0.01. The $\sigma_t$ in Eq. 11 controls the level of stochasticity is set to 0.7.

Table 5: **Hyperparameters of Training.** The batch size represents an batch size implemented via gradient accumulation.

| | SLRM | | | TAPO | |
|---|---|---|---|---|---|
| **Hyperparameter** | **SD3.5-M** | **Wan2.1-1.3B** | **Hyperparameter** | **SD3.5-M** | **Wan2.1-1.3B** |
| Training Resolution | $512 \times 512$ | $49 \times 832 \times 480$ (16 FPS) | Training Resolution | $512 \times 512$ | $81 \times 832 \times 480$ (16 FPS) |
| Learning Rate | $1 \times 10^{-5}$ | $1 \times 10^{-4}$ | Learning Rate | $1 \times 10^{-4}$ | $1 \times 10^{-4}$ |
| Training Batch Size | 32 | 16 | Training Batch Size | 8 | 8 |
| Training Epoch | 5 | 15 | Training Epoch | 5 | 10 |
| Datatype | BF16 | BF16 | Datatype | FP16 | BF16 |
| $\rho$ | $\ln 4$ | $\ln 4$ | $\beta$ | 1000 | 500 |
| | | | LoRA Rank | 64 | 128 |
| | | | Evaluation Steps $(n)$ | 8 | 8 |
| | | | SDE Latents $(N)$ | 4 | 4 |
| | | | Sampling Timesteps $(T)$ | 20 | 40 |

## B.2  VIDEO PREFERENCE DATASET COLLECTING

To evaluate the effectiveness of our method on text-to-video, we require high-quality video preference pair datasets for training the reward model SLRM. However, existing open-source datasets (Dai et al., 2024) generally suffer from low quality (short duration, poor motion coherence), collected from UNet-based models like SVD (Blattmann et al., 2023). In contrast, current video generation models are predominantly based on DiT architectures with relatively better generation quality.

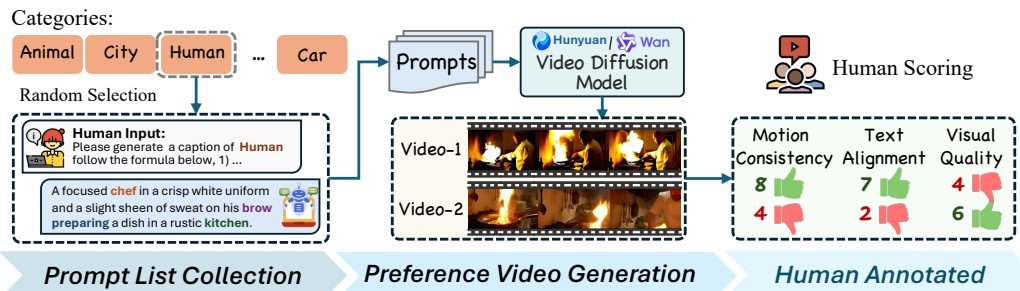

Figure 7: **Video Preference Data Collection Pipeline.**

We constructed a pipeline for collecting paired video datasets and obtained a preference-annotated dataset of 10,141 pairs through filtering. Specifically, as shown in Fig. 7, we first establish a list with 8 meta elements. Subsequently, we use LLM to extend the element category information into prompts for specific scenarios, ultimately obtaining a prompt list of 10.1k items. And the distrubution are shown in Fig. 8.

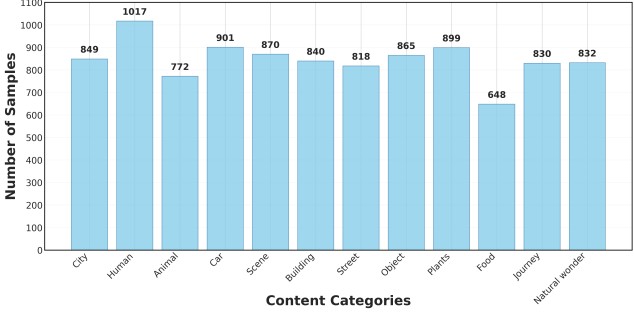

Figure 8: **Distribution of Video Preference Dataset.**

We use state-of-the-art open-source video generation models, Wan2.1-14B (Wan et al., 2025) and Hunyuan-13B (Kong et al., 2024), to generate 2 videos for each prompt. To annotate the preference order of these videos with finer granularity, we follow Flow-DPO and establish three annotation dimensions: Visual Quality (VQ), Motion Consistency (MC), and Text Alignment (TA), with human annotators scoring across these three dimensions. The annotation page can be seen in Fig. 13.

### B.3 DETAILS OF STOCHASTIC DIFFERENTIAL EQUATIONS SAMPLING

TAPO is a online training method that requires stochastic sampling. However, the diffusion model that TAPO use is based on flow matching, which relies on a deterministic generative process based on ODEs. We follow (Liu et al., 2025b) converting the deterministic ODE sampling into SDE sampling and applying it at the selected timesteps $W_T$. Here we further elaborate on this sampling process in detail.

Specifically, for a deterministic probability flow ODE of the reverse process (Song et al., 2020), it takes the following form:

$$\mathrm{d}\boldsymbol{x}_t = [f(\boldsymbol{x}_t, t) - \frac{1}{2}g^2(t)\nabla_{\boldsymbol{x}_t}\log p_t(\boldsymbol{x}_t)]\mathrm{d}t, \tag{14}$$

where $f(\boldsymbol{x}_t, t)$ denotes the drift coefficient while the $g(t)$ denotes diffusion coefficient. The $p_t(\boldsymbol{x}_t)$ represents the distribution of $x_t$ in reverse process. According to the Fokker–Planck equation (Risken, 1989), the aforementioned ODE and this probability flow SDE have the same marginal probability density:

$$\mathrm{d}\boldsymbol{x}_t = [f(\boldsymbol{x}_t, t) - g^2(t)\nabla_{\boldsymbol{x}_t}\log p_t(\boldsymbol{x}_t)]\mathrm{d}t + g(t)\mathrm{d}\mathbf{w}, \tag{15}$$

In the above equation, $g(t)$ can be expressed as the standard deviation $\sigma_t$. And according to the definition of the standard Wiener process, $\mathrm{d}\mathbf{w} = \sqrt{\mathrm{d}t}\epsilon$, where $\epsilon \sim \mathcal{N}(0, \boldsymbol{I})$. Note that flow models define a continuous-time normalizing flow through an ODE:

$$\mathrm{d}\boldsymbol{x}_t = \boldsymbol{v}_t\mathrm{d}t \tag{16}$$

Based on this special case of Eq. 14, we have:

$$\boldsymbol{v}_t = f(\boldsymbol{x}_t, t) - \frac{1}{2}g^2(t)\nabla_{\boldsymbol{x}_t}\log p_t(\boldsymbol{x}_t). \tag{17}$$

Substituting into Eq. 15, we obtain:

$$\mathrm{d}\boldsymbol{x}_t = [\boldsymbol{v}_t - \frac{\sigma_t^2}{2}\nabla_{\boldsymbol{x}_t}\log p_t(\boldsymbol{x}_t)]\mathrm{d}t + \sigma_t\sqrt{\mathrm{d}t}\,\epsilon, \tag{18}$$

The key of the equation is to establish the relationship between the score function $\nabla_{\boldsymbol{x}_t}\log q_t(\boldsymbol{x}_t)$ and the velocity field $\boldsymbol{v}_t$. Following (Liu et al., 2025b), by leveraging the linear interpolation pathand conditional expectation of $\mathbb{E}[\boldsymbol{x}_1|\boldsymbol{x}_t]$, we derive their connection through the marginal score computation. Therefore, the score function is represented as:

$$\nabla\log_{\boldsymbol{x}_t} p_t(\boldsymbol{x}_t) = -\frac{\boldsymbol{x}_t}{t} - \frac{1-t}{t}\boldsymbol{v}_t. \tag{19}$$

Substituting into Eq. 18, we have the final SDE:

$$\mathrm{d}\boldsymbol{x}_t = \left[\boldsymbol{v}_t + \frac{\sigma_t^2}{2t}(\boldsymbol{x}_t + (1-t)\boldsymbol{v}_t)\right]\mathrm{d}t + \sigma_t\sqrt{\mathrm{d}t}\,\epsilon. \tag{20}$$

Applying the Euler-Maruyama discretization for SDE and the prediction velocity $\boldsymbol{v}_\theta(\boldsymbol{x}_t, t)$ for $\boldsymbol{v}_t$ can yields our final SDE sampling scheme:

$$\boldsymbol{x}_{t+\Delta t} = \boldsymbol{x}_t + \left[\boldsymbol{v}_\theta(\boldsymbol{x}_t, t) + \frac{\sigma_t^2}{2t}(\boldsymbol{x}_t + (1-t)\boldsymbol{v}_\theta(\boldsymbol{x}_t, t))\right]\Delta t + \sigma_t\sqrt{\Delta t}\epsilon \tag{21}$$

In implementation, we use a specific noise scheduler with timestep $t$ distributed according to a logit-normal distribution (Esser et al., 2024) sampling over $[0, T]$, resulting in the following sampling scheme:

$$\boldsymbol{x}_t = \boldsymbol{x}_{t+1} - \left[\boldsymbol{v}_\theta(\boldsymbol{x}_{t+1}, t) + \frac{\sigma_t^2}{2t}(\boldsymbol{x}_{t+1} + (1-t)\boldsymbol{v}_\theta(\boldsymbol{x}_{t+1}, t))\right]\phi(t) + \sigma_t\sqrt{\phi(t)}\epsilon \tag{22}$$

where the $\phi(t)$ denotes the $\Delta t$ determined by the noise scheduler.

### B.4 TRAINING ALGORITHM OF TAPO

The algorithmic procedure of TAPO is presented in Algorithm 1.

---

**Algorithm 1** Trajectory Advantagess Preference Optimization

---

**Require:** Initial diffusion model $v_\theta$; Socre-based Latent Reward model $S(\cdot)$; prompt dataset $\mathcal{Y}$; total sampling steps $T$; SDE sampling steps $W_T = \{\tau_1, \tau_2, \ldots, \tau_n\}$
1: **for** training iteration $k = 1$ **to** $K$ **do**
2:     Sample batch prompts $y_b \sim \mathcal{Y}$
3:     **for** each prompt $y \in y_b$ **do**
4:         Init the same noise $x_1 \sim \mathcal{N}(0, \mathbf{I})$
5:         **for** sampling timestep $t = 0$ **to** $T - 1$ **do**
6:             **if** $t \in W_T$ **then**
7:                 Use SDE Sampling in Eq. 11 to get win candidates $\mathbb{X}_t^w$ and lose candidates $\mathbb{X}_t^l$.
8:                 Calculate Reward $\{s_{t,(i)}^w\}_{i=1}^P, \{s_{t,(i)}^l\}_{i=1}^P$ and select the best and worst samples $x_{t-1}^w, x_{t-1}^l$ in Eq. 12
9:             **else**
10:                 Use ODE Sampling to get $x_{t-1}^w$ and $x_{t-1}^l$ of tow branches.
11:             **end if**
12:         **end for**
13:         Obtain win-lose trajectory latents $\{x_{\tau_1}^w, x_{\tau_1}^l, x_{\tau_2}^w, x_{\tau_2}^l, \ldots, x_{\tau_n}^w, x_{\tau_n}^l\}$
14:         Computing Loss $\mathcal{L}_{TAPO}$ in Eq. 13
15:         Update diffusion model via gradient ascent: $\theta \leftarrow \theta + \eta \nabla_\theta \mathcal{L}_{TAPO}$
16:     **end for**
17: **end for**

---

## C EXTENDED EXPERIMENTAL RESULTS

### C.1 USER STUDY

We provide more details on our user study implementation. Besides qualitative and quantitative comparisons, we also conduct a user study to determine whether our method is preferred by humans. We invite 13 participants from different social backgrounds and each test session lasts about 30 minutes. During the investigation, we conducted a pairwise comparison between our method and competitors across three key dimensions: 1) Visual Quality, 2) Text Alignment, 3) Aesthetic Preference. For "Visual Quality", users were asked to select which of the two images better fine-grained details and layout quality. For "Text Alignment", users evaluated which image more accurately reflected the target text description. For "Aesthetic Preference", users judged which image aligned better with their aesthetic preferences, considering factors such as visual quality and the absence of artifacts or distortions. This comprehensive evaluation framework ensures a thorough and objective assessment of our method's performance relative to existing approaches.

The results are as shown in Fig. 9, our method defeats all competitors in all dimensions, especially in Aesthetic Preference. This highlights the powerful ability of our framework in improving more aspects beyond text-image alignment.

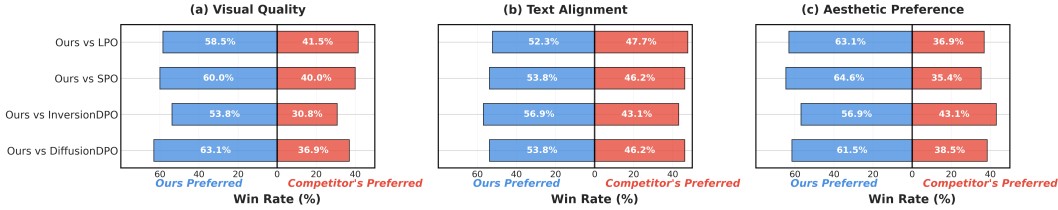

Figure 9: **User study.** The percentages indicate the proportion of users who thinks our method wins the competitor.

## C.2 MORE BENCHMARK EVALUATION

**TAPO on T2ICompBench++ (T2I).** Our method, TAPO, demonstrates state-of-the-art performance across most evaluation dimensions. Notably, TAPO achieves the highest scores in crucial areas such as Color (0.7837), Shape (0.5684), Texture (0.7036), and shows a particularly significant improvement in 2D-Spatial understanding (0.2713). This consistently superior performance compared to existing alignment models highlights TAPO's effectiveness in enhancing the text-image alignment and overall generation quality, especially for complex visual attributes and spatial arrangements.

**TAPO on Vench (T2V).** To quantify the improvement of TAPO on text-to-video generation, we compare against two baseline methods Hunyuan-13B and Wan2.1-14B, as well as one preference alignment method VideoDPO (Liu et al., 2025d). From the experimental results in Tab.8, TAPO achieves competitive performance with an overall score of 84.87, marginally outperforming VideoDPO (84.70) and other methods. TAPO demonstrates clear advantages in the key dimensions: Subject Consistency (98.79), which is highly sensitive to minor degradations that significantly impact overall quality. Notably, TAPO shows substantial improvements in Spatial Relationship (+4.62), indicating better understanding of complex spatial semantics in video generation.

**SLRM on GenAI-Bench(T2V).** To evaluate the performance of SLRM on text-to-video and validate its noise compatibility across different timesteps, we conduct experiments on GenAI-Bench and compare against existing video reward models including LiFT and VisionReward. From the experimental results, SLRM demonstrates superior performance in noisy latent evaluation.

Table 6: **Results of SLRM on GenAI-Bench (Text-to-Video).** "w/ Ties" indicates that takes account of tied pairs when calculating accuracy.

| Method | Clean Video | | $t \in [1, 500]$ | | $t \in [501, 1000]$ | |
|---|---|---|---|---|---|---|
| | w/ Ties | w/o Ties | w/ Ties | w/o Ties | w/ Ties | w/o Ties |
| LiFT (Wang et al., 2024b) | 37.06 | 58.39 | - | - | - | - |
| VisionRewrd (Xu et al., 2024b) | **51.38** | **71.04** | 43.09 | 54.81 | 42.51 | 54.07 |
| **SLRM (Ours)** | 50.66 | 64.44 | **53.30** | **67.81** | **49.28** | **62.69** |

While VisionReward achieves the highest scores on clean videos (51.38 w/ Ties), SLRM consistently outperforms all baselines across intermediate timesteps, achieving 53.38 vs. 43.09 in early timesteps t [1, 500] and 49.28 vs. 42.51 in later timesteps t [501, 1000]. These consistent performance gains validate that our score enhancement mechanism effectively preserves noise compatibility during preference learning, enabling reliable evaluation of intermediate latents throughout the diffusion process.

## C.3 SENSITIVITY ANALYSIS OF REGULARIZATION HYPERPARAMETER OF $\beta$

To investigate the impact of the regularization hyperparameter $\beta$ in Eq.13 on our method, we conduct hyperparameter analysis with results shown in Fig.10. The results demonstrate that appropriate regularization coefficients can prevent catastrophic forgetting and severe performance degradation. As illustrated in the figure, extremely small $\beta = 20$) lead to suboptimal performance across all metrics, with PickScore of 21.070, GenEval of 60.210, and HPSv2.1 of 27.050, indicating insufficient regularization that may cause the model "Catastrophic forgetting" and degrade. Conversely, excessively large $\beta = 5000$) also result in performance drops, particularly evident in PickScore (21.810) and GenEval (63.700), suggesting over-regularization that constrains optimization effectiveness. The optimal performance is achieved at moderate values, with $\beta = 500$ yielding the highest PickScore (23.210) and $\beta = 1000$ achieving peak performance on GenEval (68.830) and HPSv2.1 (32.010). Therefore, we choose $\beta = 1000$ for other experiments.

## C.4 ABLATION OF EVALUATION STEPS AND CANDIDATE LATENTS

In TAPO, the two hyperparameters governing stochastic exploration are the number of SDE sampling steps $n$, (i.e., the size of $W_T$), and the number of candidate latents, $P$, used to form win-lose pair after each SDE sampling step (i.e., the size of $\mathbb{X}_t^l = \{x_{t,(i)}^l\}_{i=1}^P$ and $\mathbb{X}_t^w = \{x_{t,(i)}^w\}_{i=1}^P$). To investigate the influence of these two hyperparameters on the quality of generated samples and computational efficiency, we conduct a detailed ablation study. Higher step numbers and larger candidate sizes mean that the alignment phase can obtain higher-quality training samples based on SLRM's prior.

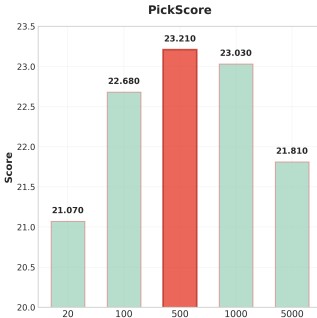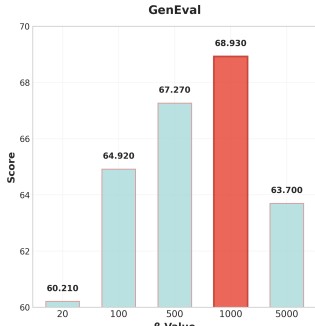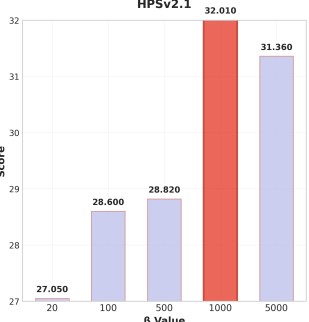

Figure 10: **Sensitivity Analysis on the hyperparameter** $\beta$ **in** $\mathcal{L}_{TAPO}$ **of Eq. 13**

Table 7: **Result of TAPO on T2I-CompBench++ (Huang et al., 2025).**

| Method | | Color | Shape | Texture | 2D-Spatial | 3D-Spatial | Numeracy | Non-Spatial | Complex |
|---|---|---|---|---|---|---|---|---|---|
| **Base Model** | SD-XL | 0.5592 | 0.4230 | 0.5172 | 0.2009 | 0.3172 | 0.4631 | 0.3105 | 0.3409 |
| | SD3.5-M | 0.6810 | 0.4921 | 0.6295 | 0.2293 | 0.3491 | 0.5137 | 0.3108 | 0.3513 |
| | FLUX.1 Dev | 0.6971 | 0.5130 | 0.6123 | 0.2503 | 0.3416 | 0.5246 | 0.3127 | 0.3679 |
| **Alignment Model** | Diffusion-DPO | 0.6829 | 0.5190 | 0.6338 | 0.2322 | 0.3620 | 0.5295 | 0.3155 | 0.3663 |
| | Inversion-DPO | 0.6910 | 0.5189 | 0.6405 | 0.2461 | 0.3598 | 0.5402 | 0.3161 | 0.3708 |
| | SPO | 0.7296 | 0.5392 | 0.6762 | 0.2409 | 0.3703 | 0.5724 | 0.3127 | 0.3721 |
| | LPO | 0.7460 | 0.5508 | 0.6793 | 0.2541 | 0.3822 | **0.5835** | 0.3158 | 0.3838 |
| | **TAPO** | **0.7837** | **0.5684** | **0.7036** | **0.2713** | **0.4013** | 0.5794 | **0.3217** | **0.3961** |

As depicted in Fig. 11, for moderate values of $P$ (e.g., $P = 2, 3, 4$), increasing the SDE steps n (from 1 to 16) generally leads to improvements in visual quality and prompt alignment. However, our study reveals that this benefit does not extend indefinitely. When these hyperparameters becomes excessively large (e.g., P=5), this may lead to reward hackingmeaning the model overfits the reward signal by generating 'win' samples that are superficially preferred but lack genuine quality. This suggests that while increased stochastic exploration can enhance sample quality, an overemphasis on it can cause the reward model (SLRM) to exploit spurious patterns or artifacts in its reward landscape, resulting in visually unappealing outputs that paradoxically achieve high reward scores. More results can be seen in Fig. 16 and Fig. 17

## C.5 ANALYSIS OF REWARD INCONSISTENCY DURING SAMPLING

As discussed in the introduction, different timesteps in diffusion models emphasize distinct aspects, resulting in inconsistent reward signals. To intuitively demonstrate this problem, we present two cases from the trained model's sampling trajectories to illustrate this phenomenon. As shown in Fig 12, in the first case, despite its overall higher quality, the 'win' sample's composition and layout are less coherent than the 'lose' sample in early stages ($t = 850, 750$), leading to a lower reward

Table 8: **Result of TAPO on VBench (Huang et al., 2024).** We apply our TAPO on Text-to-Video (T2V). "§" indicates the VideoDPO (Liu et al., 2025d) conduct on our Preference Video Dataset (Appendix B.2). VBench consists of 16 dimensions, and we present several key dimensions that measure video quality and semantics, along with the overall score of other dimensions. "SC": Subject Consistency, "AQ": Aesthetic Quality, "MS": Motion Smoothness, "OC": Object Class, "Human Action", "SR": Spatial Relationship.

| Method | Quality Score | | | Semantic Score | | | Overall Score | | |
|---|---|---|---|---|---|---|---|---|---|
| | SC | AQ | MS | OC | HA | SR | Quality | Semantic | Total |
| Hunyuan-13B (Kong et al., 2024) | 97.37 | 60.36 | 98.99 | 86.10 | 94.40 | 68.68 | 85.09 | 75.82 | 83.24 |
| Wan2.1-14B (Wan et al., 2025) | 97.52 | 66.07 | 98.30 | 86.28 | 95.40 | 75.39 | 85.59 | 76.11 | 83.69 |
| Wan2.1-1.3B (baseline) | 96.34 | 62.43 | 97.44 | 88.81 | **98.20** | 76.46 | **85.30** | 80.09 | 84.26 |
| VideoDPO § (Liu et al., 2025d) | 96.68 | 64.80 | 98.10 | 90.26 | 96.64 | 80.25 | 85.00 | 80.95 | 84.70 |
| TAPO | **98.79** | **67.27** | **98.12** | **89.62** | 98.00 | **81.08** | 85.21 | **82.49** | **84.87** |

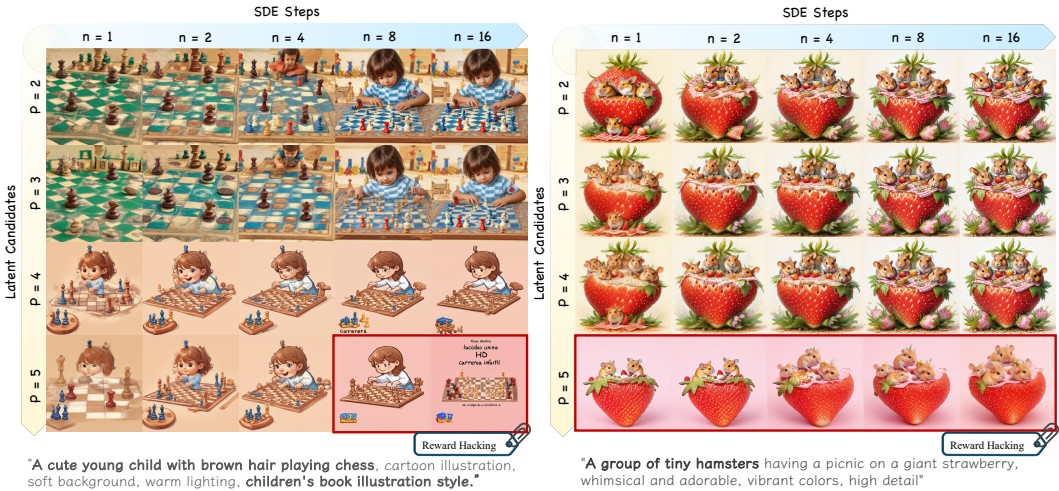

Figure 11: **Results under Different evaluation Steps $n$ and Latent Candidates Size $P$.** "Reward hacking" is observed with an excessive number of evaluation steps and large latent candidate sizes.

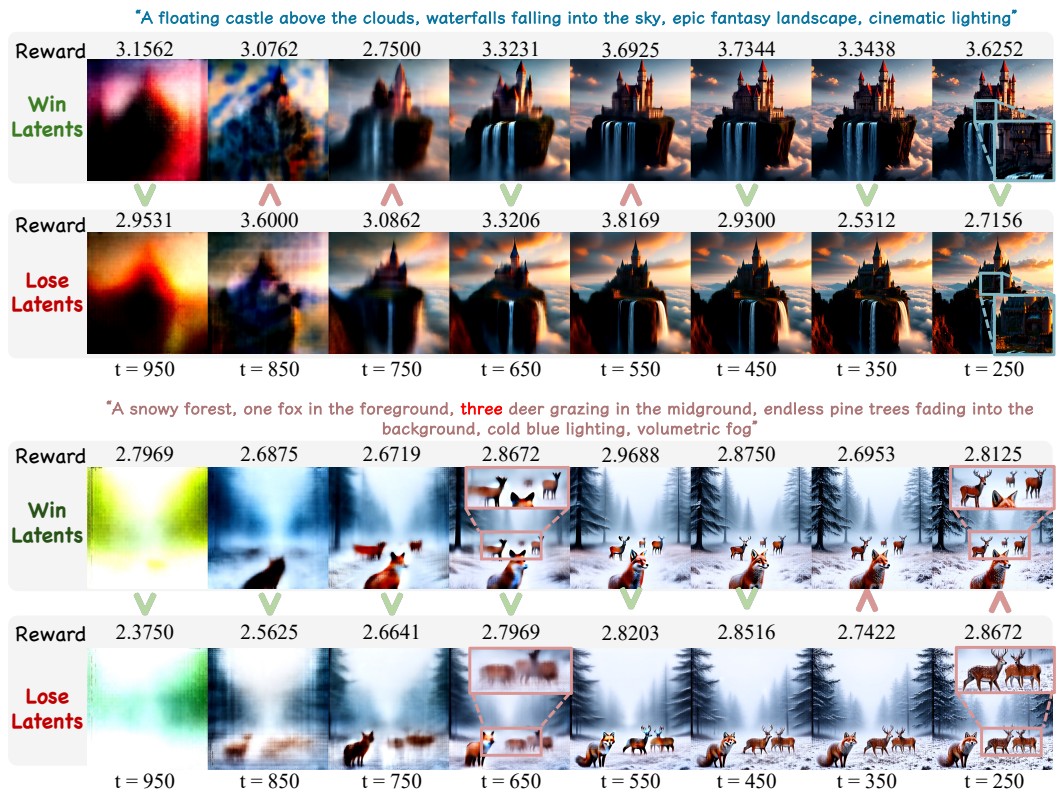

Figure 12: **Inconsistent Reward Across Denoising Timesteps.** The critical challenge of inconsistent reward signals for noisy latents across different timesteps, a core motivation for our work.

score. In contrast, in later steps ($t = 250$), it achieves a higher reward score due to its refined details. A similar inconsistency is observed in Case 2. For instance, at an early timestep ($t = 650$), the 'win' sample better captures the concept of "three deer," thus getting a higher score.

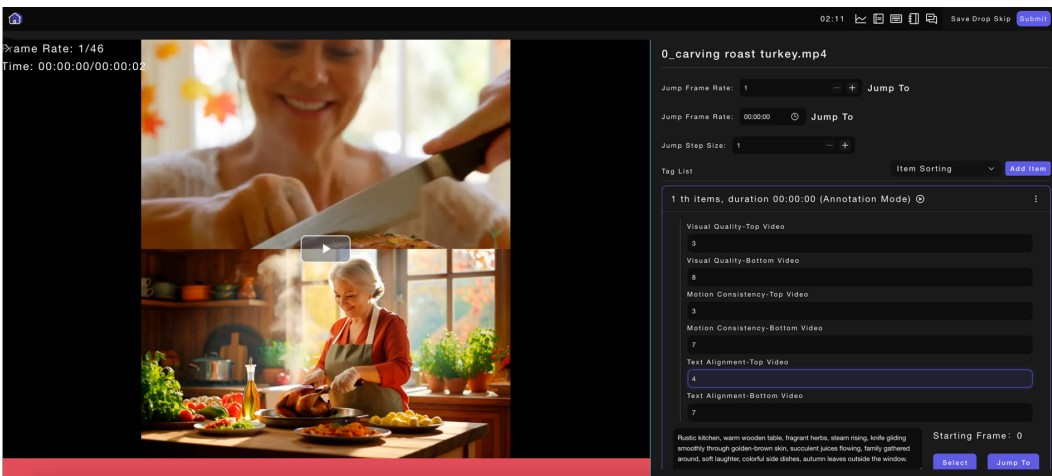

Figure 13: **Video Preference Dataset Annotation Interface.**

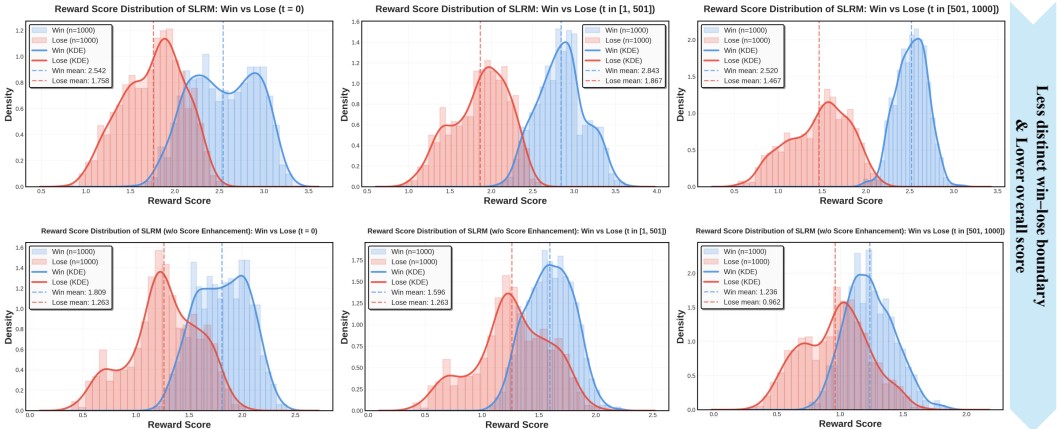

Figure 14: **Visualization about the Influence of Score Enhancement on Reward Distributions.**
'w/o Score Enhancement' exhibits lower overall reward scores, and the separation between win and lose latents becomes significantly blurred when $t \neq 0$, indicating reduced discriminability under noisy latent conditions.

### C.6 ANALYSIS OF THE SCORE ENHANCEMENT

To more clearly demonstrate the influence of the score enhancement mechanism on the reward score $\hat{S}(x_t, c)$ produced by SLRM, we visualize the reward distributions of 1,000 sample pairs from the Pick-a-Pic v1 test set. For each pair, we plot the distributions of $\hat{S}(x_t^w, c)$ and $\hat{S}(x_t^l, c)$ across different diffusion timesteps. As shown in Fig. 14, with score enhancement, the win–lose reward distributions remain well separated across all timesteps, and the overall reward magnitude stays high. This indicates that SLRM can consistently discriminate between high-quality and low-quality latents, even under noisy conditions (i.e., t=0). In contrast, without score enhancement, the reward distributions shift toward lower values, and the separation between win and lose latents becomes significantly blurred as the timestep increases. This suggests that the model struggles to maintain reliable reward predictions when operating on noisy latents. Overall, these results show that score enhancement substantially improves the stability and discriminability of SLRM under noisy latents, validating its necessity for robust reward modeling throughout the diffusion trajectory.

Figure 15: **Visualization about the Influence of "Training on $t = 0$ Only" on Reward Distributions.**

# D MORE VISUALIZATION

We present additional experimental visualization of our TAPO, including text-to-image in Fig. 18 and text-to-video in Fig. 19, which demonstrate that our method outperforms existing approaches in human aesthetic preference, text alignment, and other aspects.

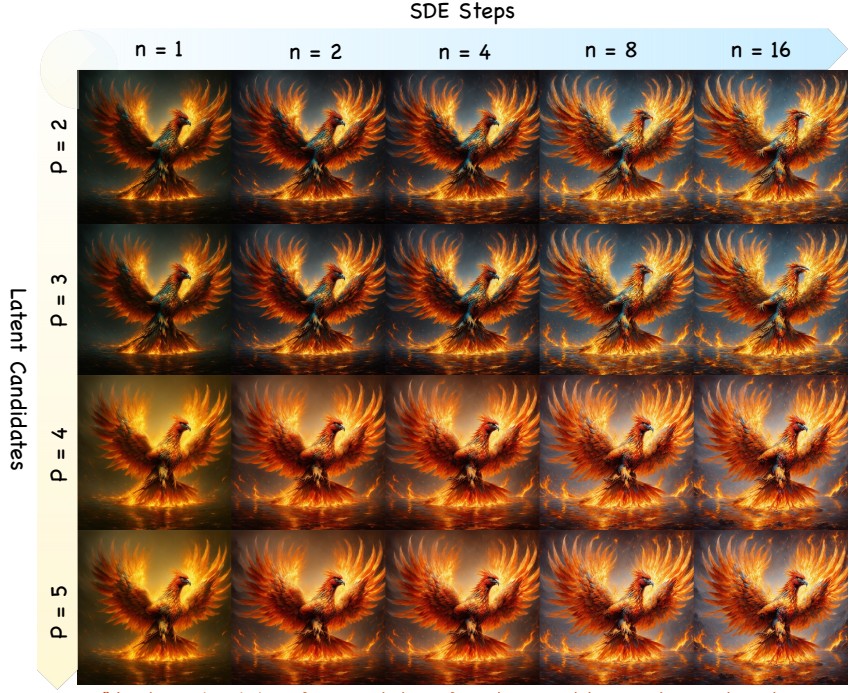

Figure 16: **Results under Different SDE Steps $n$ and Latent Candidates Size $P$.**

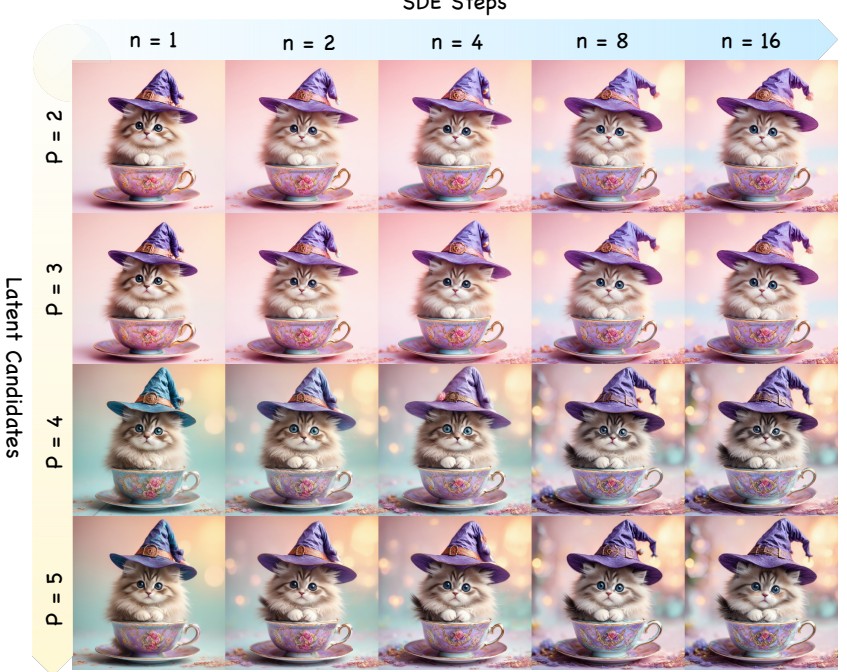

Figure 17: **Results under Different SDE Steps $n$ and Latent Candidates Size $P$.**

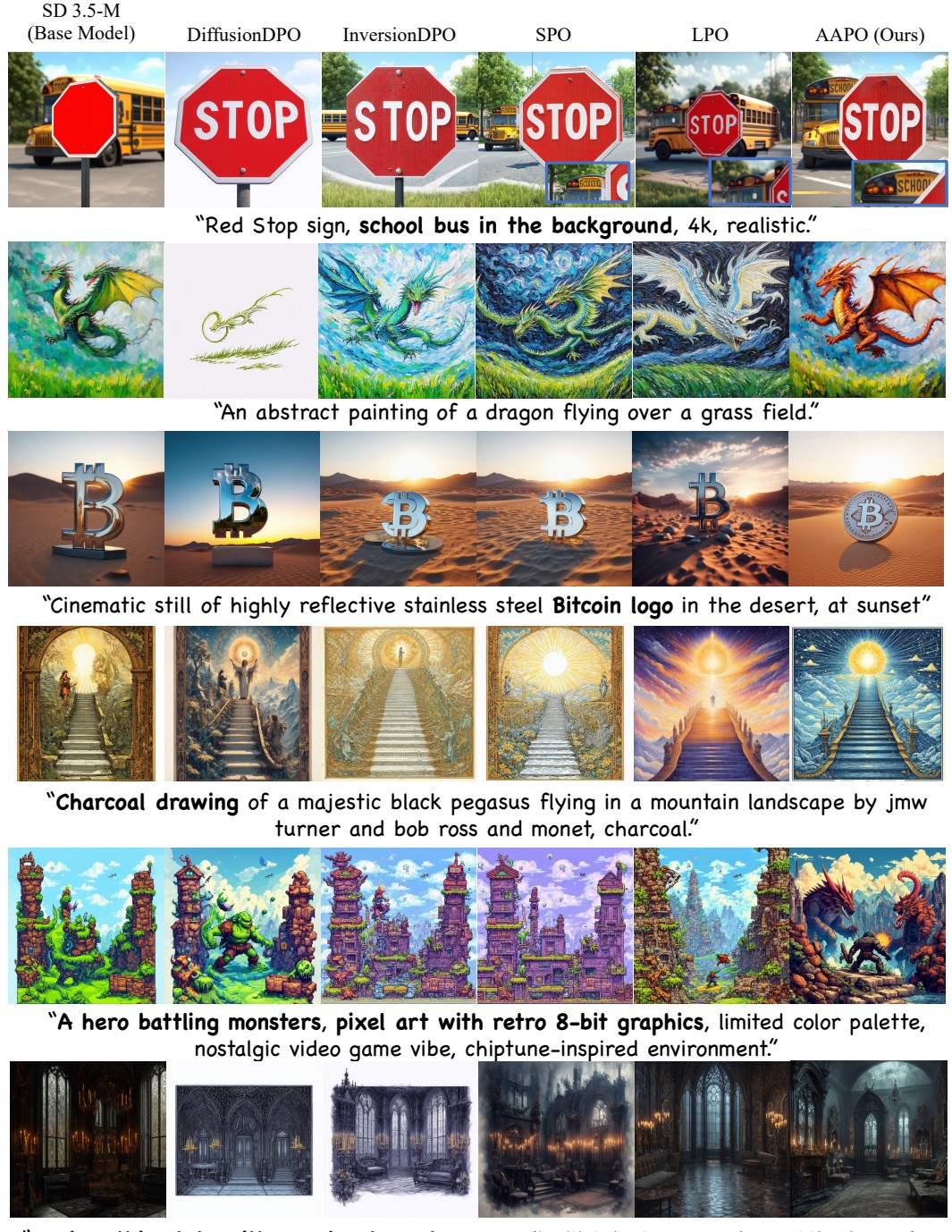

Figure 18: **Qualitative Comparison of Preference Pptimization Methods.** Rows 1-3 show the alignment of the subjects, and rows 4-6 show the alignment of style.

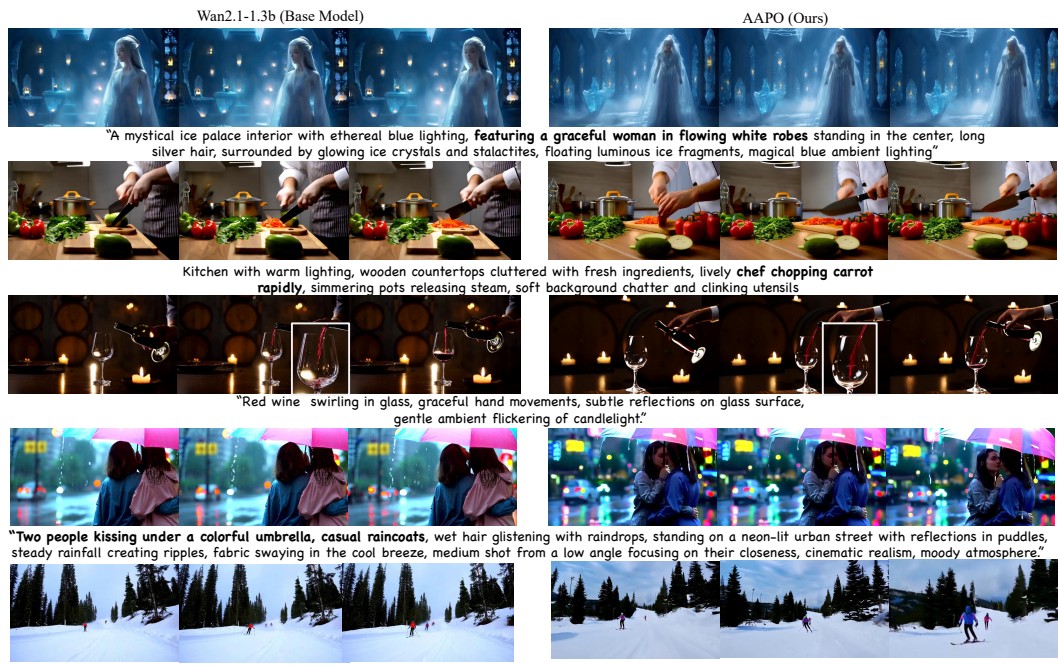

Figure 19: **Results of TAPO in T2V.**

