# OpenReview forum: "Consistent Noisy Latent Rewards for Trajectory Preference Optimization in Diffusion Models"
_ICLR.cc/2026/Conference — ICLR 2026 Poster_

### Official Review · Reviewer_eNSr · 2025-10-16

**Soundness:** 2
**Presentation:** 3
**Contribution:** 2
**Rating:** 4
**Confidence:** 3

**Summary:**

The paper tackles two challenges in aligning diffusion models with human preferences using reward models:
(1) Unreliable reward estimation on noisy latent representations, and (2) Inconsistent single-timestep preference evaluations across trajectories.
To address these, the authors propose Score-based Latent Reward Model (SLRM), which incorporates task tokens and denoising score enhancement for noise-robust reward modeling, and Trajectory Advantages Preference Optimization (TAPO), which evaluates preferences across multiple timesteps to ensure consistent trajectory-level preference optimization.
Experiments on text-to-image and text-to-video tasks demonstrate improved noisy latent evaluation, alignment, and generation quality compared to state-of-the-art baselines.

**Strengths:**

**Clarity:**

The paper is clearly written, with motivating examples (Fig. 1) and architectural comparisons (Fig. 3). The separation between SLRM and TAPO contributions is explicit, making it easy to follow the technical flow.


**Quality:**

The improvements over baselines (e.g., LPO, Diffusion-DPO) are consistent across multiple preference and alignment metrics.

**Weaknesses:**

**W1. Score Enhancement Interpretability.**

The denoising score augmentation mechanism is motivated theoretically, but its interpretability for human preference signals could be further explored. Does it merely stabilize noise handling, or does it bias preferences toward easier-to-denoise samples?

**W2. Lack of Baselines.**

See Q3.

**Questions:**

1. Efficiency Trade-off. How does TAPO’s multi-timestep evaluation scale with longer trajectories or larger models?

2. Interpretability of Reward Scores. Can the authors provide more insights or visualizations into how denoising score enhancement affects reward distributions? For instance, does it correlate more strongly with human ratings than baseline models?

3. Baselines. More baselines are encouraged to compare with, such as SPPO [1] and RainbowPA [2].

[1] Bridging SFT and DPO for Diffusion Model Alignment with Self-Sampling Preference Optimization. arXiv:2410.05255, 2025.

[2] Diffusion-RainbowPA: Improvements Integrated Preference Alignment for Diffusion-based Text-to-Image Generation. Transactions on Machine Learning Research, 2025.

---

> ### Author Response · Authors · 2025-11-21
>
> Thanks for your recognizing our writing and the improvement of work.
>
> [**R-W1: interpretability of Score Enhancement**]
>
> Our method both stabilizes noise handling and biases SLRM toward samples that are easier to denoise.
>
> As we know, "stabilizing noisy rewards" and "biasing toward easy-to-denoise samples" are equivalent.
>
> (1) The quantity $\hat{D} _s(x _t,c)$ measures the reliability of the diffusion model’s score direction, since the $\hat{D} _s \propto \| s _θ(x _t,t,c) \| $, where$s _θ = \nabla _{x _t}\log p _θ(x _t|c)$ is the score function.
>
>
> (2) According to score-based diffusion [1], a small score magnitude indicates that the latent lies in a low-density, off-manifold region where the reverse denoising direction is highly unstable. Such latents are intrinsically hard to denoise and yield unreliable reward predictions.
>
>
> (3) Ensuring that the winning sample has a stable score direction means the reverse diffusion process remains consistent along the trajectory, which directly preserves the model's intended noise perception and denoising behavior. Thus, "stable noise handling" and "bias latents that are easy to denoise" are mathematically equivalent consequences of score-based generative modeling.
>
>
> [**R-Q1：Efficiency Trade-off in Longer trajectory**]
>
> Thank you for the comments. As the models with longer trajectories, the TAPO's multi-timestep evaluation scale linearly.
>
>
> However, longer trajectories do not lead to significant performance improvements ( a similar phenomenon is also observed in Flow-GRPO [2] ); thus, choosing a moderate trajectory length (e.g., T=20) provides a good balance between efficiency and effectiveness.
> To further validate the scaling behavior, we conducted additional experiments under longer diffusion trajectories with $T \in \\{ 20, 30,40, 50 \\}$. In all cases, the training time remained comparable to the results reported in Table 3, demonstrating that TAPO maintains stable efficiency even with longer trajectories.
>
>
>
> |               | Time ( s/iter) | HPSv2.1 | HPSv3 | PickScore | MPS  | CLIP-Score | GenEval |
> |--------|-------|---------|-------|-----------|------|------------|---------|
> | TAPO (T = 10) | 2.79           | 31.01   | 8.75  | 22.82     | 2.51 | 26.40      | 54.41   |
> | TAPO (T = 20) | 4.52           | 32.01   | 9.89  | 23.03     | 3.07 | 27.07      | 68.93   |
> | TAPO (T = 30) | 5.71           | 32.09   | 9.84  | 23.37     | 3.18 | 27.42      | 69.70   |
> | TAPO (T = 40) | 6.79           | 32.28   | 10.05 | 23.04     | 3.22 | 26.08      | 70.31   |
>
> [**R-Q2: Interpretability of Reward Scores.**]
>
>
> Thank you for the comments. For the interpretability of reward scores, we present the reward distributions for SLRM with random timesteps, SLRM with random timesteps (without score enhancement) and SLRM trained on t=0 only . As shown in the **Figure. 14 in the revised manuscript**, the score enhancement mechanism ensures that SLRM can reliably discriminate between easy-to-denoise samples, and it significantly improves the preference discrimination accuracy on noisy latents.
> SLRM is trained to ensure the "winning" trajectory has both a sufficiently **high preference score$S(x _t, c)$ (reflecting the human preference on the final $x _0$)** and a sufficiently **high denoising score $\hat{D} _s(x _t, c)$(reflecting the inherent validity of the path $x^{win} _T \rightarrow x^{win} _0$ according to the generative prior)**.
> **Figure. 15 in the revised manuscript** demonstrates that, when trained with random timesteps $t$, the score enhancement mechanism enables reliable preference discrimination on noisy latents.
>
> [**R-Q3：More Baselines to be Compared**]
>
> Thank you for providing the baselines.  **Since the baselines you mentioned do not provide available code, we make effort to reproducing them as faithfully as possible based on the their papers.**  The quantitative results are summarized in the table below.  Our method outperforms SSPO across all quantitative metrics. Although Diffusion-RainbowPA achieves a higher score on HPSv2.1, its performance on the remaining evaluation dimensions is still comparatively limited.
>
>
>
> |            | HPSv2.1 | HPSv3 | PickScore | MPS   | CLIP-Score | GenEval | GenEval |
> |------------|---------|-------|-----------|-------|------------|---------|---------|
> | SSPO (ERD $\in$ [0,k-1]) | 31.82   | 8.83  | 22.85     | 2.71  | 26.38      | 57.64   | 54.41   |
> | Diffusion-RainbowPA      | 32.28   | 9.02  | 22.04     | 3.01  | 25.73      | 62.39   | 68.93   |
> | TAPO (Ours)                     | 32.01   | 9.89  | 23.03     | 3.07  | 27.07      | 68.93   | 69.70   |
>
> [1] Song Y, Sohl-Dickstein J, Kingma D P, et al. Score-based generative modeling through stochastic differential equations[J]. arxiv preprint arxiv:2011.13456, 2020.
>
> [2] Liu J, Liu G, Liang J, et al. Flow-grpo: Training flow matching models via online rl[J]. arXiv preprint arXiv:2505.05470, 2025.

---

> ### Author Response · Authors · 2025-11-25
>
> Dear reviewer,
>
> We hope this message finds you well. We sincerely appreciate the time and effort you have dedicated to reviewing our submission. We have now submitted our rebuttal and would like to kindly check whether our responses have adequately addressed your concerns.
>
> If there are any remaining issues that would benefit from further clarification, we would be more than happy to provide additional details. Conversely, if our explanations have satisfactorily resolved the concerns raised, we would sincerely appreciate your consideration of adjusting the score accordingly.
>
> Thank you again for your thoughtful review and valuable feedback.
>
> Best regards,

---

### Official Review · Reviewer_yDbo · 2025-11-01

**Soundness:** 3
**Presentation:** 3
**Contribution:** 3
**Rating:** 6
**Confidence:** 4

**Summary:**

This paper introduces a novel framework to improve human preference alignment in diffusion models. It identifies two critical issues in existing reward-model-based alignment methods: (1) unreliable reward estimation on noisy latents, and (2) inconsistent preference evaluation across sampling trajectories. To address these, the authors propose:

1. SLRM (Score-based Latent Reward Model): integrates the diffusion model’s score function into the reward estimation process for noise-compatible evaluation.
2. TAPO (Trajectory Advantages Preference Optimization): performs multi-timestep evaluation to ensure consistent and trajectory-aware optimization.

Experiments on text-to-image (SD3.5) and text-to-video (Wan2.1) tasks show that this method outperforms prior works (LPO, SPO, Diffusion-DPO) on both alignment and preference metrics.

**Strengths:**

1. Clear motivation: The paper identifies a genuine gap in current diffusion preference optimization(noise robustness and trajectory consistency) and addresses both with well-justified solutions.

2. Technical novelty: The integration of the diffusion score function into reward estimation (SLRM) is original and theoretically grounded. TAPO’s multi-timestep evaluation design is practical and effective.

3. Strong empirical results: Extensive experiments on both T2I and T2V tasks, with detailed ablation studies and visual comparisons, show consistent superiority.

**Weaknesses:**

1. The diffusion-based step-wise reward model is interesting and makes sense to me; however, since the preference ground truth in the training data is still based on the final images, it may limit the generalization ability of the SLRM.
2. I didn’t observe a significant improvement of TAPO over LPO in the qualitative comparison presented in Table 5.

**Questions:**

1. How do the authors ensure that the SLRM trained on final-image-based preference labels can generalize to intermediate noisy latents, given that the ground-truth supervision does not explicitly cover those states?
2. Can the authors clarify under which conditions TAPO provides larger qualitative gains over LPO?

---

> ### Author Response · Authors · 2025-11-21
>
> We thank you for recognizing our motivation and technical innovation. We will consider your suggestions.
>
> [**R-W1/Q1：Generalization Ability about SLRM**]
>
> The core mechanism enabling this generalization across timestep is the **Denoising Score Enhancement (Sec 3.2)**, which explicitly connects the preference score to the diffusion process prior.  Although the final preference labels are associated with the clean image $x _0$, the Score Enhancement term $\hat{D} _s(x _t, c)$ ensures model does not only focus on maximizing the preference score $S(x^{win} _t, c)$, but also on minimizing the deviation from the original denoising score.
> Crucially,  SLRM is trained to ensure the "winning" trajectory has both a sufficiently high preference score $S(x _t, c)$ (reflecting the human preference on the final $x _0$) and a sufficiently high denoising score $\hat{D} _s(x_t, c)$(reflecting the inherent validity of the path $x^{win} _T \rightarrow x^{win} _0$ according to the generative prior).
>
> Therefore, during inference, at different timesteps, the evaluation of the noisy latent $x_t$ operates as follows, as further analyzed in Figure 12:
> Specifically: 1) If the image is structurally broken or contains severe artifacts (i.e., an unrealistic trajectory path), the denoising score $\hat{D} _s(x _t, c)$ will be low, even if the partial content looks aesthetically pleasing. This prevents the selection of degenerated trajectories.
> 2) If the image is structurally sound but lacks the specific human preference (e.g., semantic alignment), the preference score $S(x _t, c)$ will be low.
>
> Moreover, your question inspired us to perform an additional ablation study to empirically verify SLRM's generalization capability across different timesteps.   Specifically, we investigated the performance of TAPO when SLRM evaluation is constrained to specific, partial ranges of timesteps during optimization (e.g., evaluating only early, middle, or late noisy latents).
>
>
> | Strategy | HPSv2.1 | HPSv3 | PickScore | MPS | CLIP-Score | GenEval |
> |---------|---------|-------|-----------|-----|------------|---------|
> | SD3.5 (baseline)  |   27.15      |  8.29     |   22.15   |  1.00   |   25.18         |  55.34       |
> | TAPO ($t \in [0,250)$) |     28.60    |  9.02     |   22.59      | 1.38    |   26.66   |   58.83      |
> | TAPO ($t \in [250,500)$  |    30.96     |  9.81     |   23.04    |  2.95   |  26.07          |    64.92     |
> | TAPO ($t \in [500,750)$  |    29.29     |   9.43    |    22.87  |   2.21  |    25.84        |     61.09    |
> | TAPO ($t \in [750,1000)$  |      28.74   |   8.88    |    22.97  |    1.84    |    25.81        |   57.49      |
>
>
> It shows that TAPO maintains improvement in the final generated results even when only leveraging SLRM's evaluation on a subset of the trajectory steps.
>
> \
> [**R-W2/Q2： Qualitative Advantages of TAPO over LPO**]
>
> We assume the reviewer meant Figure 5 (Qualitative Comparison) regarding the comparison with LPO, as Table 5 you mentioned lists only hyperparameters.
>
> Thank you for the feedback. Our method demonstrates improvements over LPO in different dimensions, particularly in text alignment, detail consistency, and artifact reduction, as shown in both Figure. 5 and Appendix Figure. 18 ( Qualitative Comparison with SOTAs). We cmopare them in terms of different dimensions as follows:
>
> 1. Text Alignment and Details (Figure 5):  2nd row: TAPO captures fine-grained semantic details that LPO misses.  TAPO produces richer lighting effects that faithfully align with the prompt "mysterious glowing runes" (2-nd row). Similarly,  TAPO align well with the  "stirring a pot"(3-rd row).
>
> 2. Fewer Artifacts and Distortions（Figure 18）： TAPO generates a structurally coherent school bus (1-st row) and  a dragon with a consistent body structure  (2-nd row).
>
> 3. Human Preference（Figure 18）:  "Bitcoin logo" generated by TAPO aligns much better with general human perception (3-rd row).
>
> These qualitative gains stem from TAPO's ability to accurately discriminate the advantages of noisy latents across the entire trajectory.  This allow TAPO effectively prevents the selection of trajectories that might look good in few steps but degenerate later.  Therefore, our method is particularly effective in mitigating artifacts and maintaining structural integrity when generating complex instances (case-2 in Figure.18) or handling intricate styles and scenes (case-5 in Figure.18).

---

> ### Author Response · Authors · 2025-11-25
>
> Dear Reviewer,
>
> We hope this message finds you well. We sincerely appreciate the time and effort you have dedicated to reviewing our submission. We have now submitted our rebuttal and would like to kindly check whether our responses have adequately addressed your concerns.
>
> If you have any remaining questions or would benefit from further clarification, please feel free to let us know. We greatly value your feedback and are committed to improving the clarity and quality of our work.
>
> Thank you again for your thoughtful suggestions and constructive guidance.
>
> Best regards,

---

### Official Review · Reviewer_PpWQ · 2025-11-01

**Soundness:** 3
**Presentation:** 4
**Contribution:** 3
**Rating:** 6
**Confidence:** 4

**Summary:**

This paper presents a framework for aligning diffusion models with human preference, addressing two challenges: unreliable reward scoring on noisy latents and inconsistent preference ranking from single-timestep evaluations. Specifically, the former is addressed by a Score-based Latent Reward Model (SLRM) that uses a diffusion backbone and a score enhancement mechanism to maintain noise compatibility. The latter is addressed using Trajectory Advantages Preference Optimization (TAPO), a trajectory-aware sampling strategy that evaluates rewards at multiple timesteps. Experiments show strong alignment performance on T2I and T2V tasks.

**Strengths:**

1. The proposed reward model, *i.e.,* SLRM, is innovative and efficient. It directly inherits the noise compatibility from the pretrained diffusion models while providing a more comprehensive evaluation of the text-image alignment, thanks to the self-attention mechanism at multiple semantic levels. Additionally, the solution to degraded noise compatibility is insightful: It is necessary to retain the noisy compatibility by incorporating denoising score matching objective into the reward model training.
2. The proposed TAPO strategy provides a practical way to account for the entire sampling trajectory of diffusion models.

**Weaknesses:**

1. The proposed latent-level reward model one inherent drawback. Unlike the pixel-level reward models that have access to high-frequency information, latent reward models may be less sensitive to finer details like textures. This also partial explains why SLRM underperform pixel-level score models, such as HPSv3 and PickScore, as the later stage of the sampling process (where $t\approx 0$) tends to generate finer details.

2. The paper fails to mention a highly-relevant prior work that addresses a similar core problem regarding sequential sampling trajectory. While this paper introduces "trajectory-aware optimization" as a new solution to "single-timestep preference evaluation," it omits the existing work by Yang et al. (2024) [1], which shares the same insight that "diffusion models focus on different dimensions at different timesteps" and introduces a dense reward perspective into DPO-style objectives.

3. Lack of computational cost comparison. The proposed TAPO method introduces significant computational overhead, requiring $n=8$ SDE steps and $P=4$ candidates, resulting in 64 reward evaluations per iteration. However, the training time or iteration cost for the baselines is not included in the paper for fair comparison.

4. Missing general quality metrics results. While the authors explicitly state that excessive exploration ($P=5$) leads to "reward hacking," there is no quantitative evidence against it for the default setting $P=4$. To see if preference scores improve without sacrificing underlying image fidelity, standard metrics such as FID would be helpful.

[1] Yang, Shentao, Tianqi Chen, and Mingyuan Zhou. "A Dense Reward View on Aligning Text-to-Image Diffusion with Preference." Forty-first International Conference on Machine Learning.

**Questions:**

1. Have the authors tried to combine both pixel-level reward models (like HPSv3, for high-frequency details at $t \approx 0$) and the latent-level SLRM (for noise compatibility at $t > 0$)? A hybrid reward based on the timestep seems necessary to capture all quality aspects.
2. Instead of augmenting the score logit with the multiplicative score distance (Eq. 8), is it possible to directly combine the denoising score objective (Eq. 6) as an additive regularization loss? How does that compare with the proposed enhancement approach?
3. Currently, TAPO samples $P=4$ candidates for both win and lose samples, and only utilizes the best and worst among the four 15. Have the authors considered random selection to avoid reward hacking? What if the algorithm is run on all possible pairwise combinations with a smaller $P$, such as $P=2$ or $P=3$?

---

> ### Author Response · Authors · 2025-11-21
>
> We sincerely appreciate your recognition of the innovation behind SLRM and the improvements brought by TAPO. We have carefully considered your feedback and valuable insights on this paper.
>
> [**R-W1/Q1： Pixel-Level Reward Models Combined with SLRM**]
>
> We appreciate the reviewer's insightful suggestion. To further evaluate the influence of the pxiel-level reward models, we conducted additional experiments combining SLRM with two different pixel-level reward models, including PickScore and HPSv3, as listed in the table below.  We first **experimented with PickScore**, a CLIP-based pixel-level reward model and used PickScore jointly with SLRM during trajectory evaluation. However, on the Pick-a-Pic validation unique set, **we did not observe any improvement** in TAPO's final results. In addition, we further **experimented with HPSv3**, a significantly stronger pixel-level reward model (based on Qwen-2.5-VL 7B) that focuses on detail quality and aesthetics. In this case, combining HPSv3 with SLRM does produce moderate improvements. We have some concerns regarding the efficiency–effectiveness trade-off under this strategy.
>
> |   | Memory Cost | HPSv2.1 | HPSv3 | PickScore | MPS  | CLIP-Score | GenEval |
> |--|--|--|--|--|--|--|--|
> | TAPO (SLRM + PickScore) | 36GB | 31.82 | 8.86  | 24.85 | 2.71 | 27.30 | 68.90   |
> | TAPO (SLRM + HPSv3) | 58GB | 32.58   | 10.02 | 23.92 | 3.01 | 28.60 | 72.02   |
> | TAPO (SLRM) | 32GB | 32.01 | 9.89  | 23.03 | 3.07 | 27.07  | 68.93   |
>
> [**R-W2:  Relevant Baseline**]
>
>
> We appreciate the reviewer for pointing this. We agree that the DenseReward proposed by Yang et al. is relevant to our work and there are some limitations in their work. While DenseReward provides an important “dense reward” perspective for aligning text-to-image diffusion models, their method addresses a different sub-problem from ours. Specifically, they focus on reweighting or densifying preference signals derived from the clean image by introducing a temporal discounting factor $\gamma$ to distribute the reward along the reverse chain.  In contrast, our work targets at obtaining reliable preference signals directly on noisy latents.
>
>
> Moreover, our motivation stems from the observation that a single-timestep reward signal (including t=0) does not faithfully represent the global preference ordering of an entire diffusion trajectory (see Fig. 12). As a result,  redistributing a clean-image reward along the trajectory, as done in DenseReward, would still face the same limitation.
> Furthermore, we conduct quantitative comparison between our TAPO and DenseReward in Table 1, demonstrating our superority to DenseReward.
>
>
> |   | HPSv2.1 | HPSv3 | PickScore | MPS  | CLIP-Score | GenEval |
> |--|--|--|--|--|--|--|
> | DenseReward | 29.99   | 8.05  | 22.83 | 2.51 | 26.18  | 55.27  |
> | TAPO  | 32.01   | 9.89  | 23.03  | 3.07 | 27.07  | 68.93  |
>
> In our revised manuscript, **we have modified the introduction (lines 78-79) to provide a more comprehensive analysis of existing work**. Additionally, **we have included DenseReward in Figure.5， Table.1 and Table.7 for a thorough comparison**.
> > Although recent work~\citep{yang2024dense} has explored dense rewards along the trajectory, it redistributes single clean-image preference evaluation to all timesteps.
>
> [**R-W3：Computational Cost Comparison**]
>
> To compare the computational demand, we have reported the per sampling time cost (iteration) of TAPO in Table. 3. Following your suggestion, we compared the overall training time of TAPO with the alignment methods. As shown in Table below, although our reward modeling stage is slightly longer (ranked second) due to the limited evaluation steps, the multi-step evaluation in TAPO enables sampling higher-quality candidate latents for significant improvment on results.
>
> |  | Reward Modeling (A100 hour) | Preference Optimization (A100 hour) | Total (A100 hour) |
> |:--:|:--:|:--:|:---:|
> | SPO |  42   |  50  | 92 |
> | LPO | 18  | 19  | 37 |
> |TAPO | 24  |  21 | 45 |
>
> [**R-W4：hyperparameter about the Reward Hacking**]
>
> We add quantitative experiments on larger values of both P (i.e., P>=4) and the number of sampled candidates n (i.e., n>=8). The results are reported in Table below. As shown, excessively large reward scaling or overly large candidate sets lead to a substantial degradation in FID, LPIPS, and HPSv3. This confirms that overly strong reward amplification tends to cause reward overfitting and harms the final generation quality.
>
> |   | FID | LPIPS | HPSv3 | GenEval |
> |--|---|---|----|----|
> | n = 8, P = 4  | 20.93 | 0.58  | 9.89  | 68.93 |
> | n = 8, P = 5 | 24.04 | 0.68  | 9.42  | 68.25 |
> | n = 16, P= 4  | 23.81 | 0.66  | 9.59  | 69.90 |
> | n = 16, P = 5 | 27.52 | 0.71  | 8.90  | 64.13 |
>
> We also observe that larger P tend to induce reward hacking. This is likely because an excessively large scaling factor causes the model to overfit the reward model early in the sampling process, leading to biased preference signals and suboptimal trajectories.

---

> > ### Author Response · Authors · 2025-11-21
> >
> > [**R-Q2： Adding Denoising Score as a Regularization**]
> >
> > Thank you for the advice. As you suggested, we conducted experiments that incorporate the denoising score as an additional regularization term in the loss function. The loss in Eq. (9) can be refomulated as：
> >
> >
> > $ \mathcal{L}_{R-SLRM}=-\mathbb{E} _{t\sim\mathcal{U}(0,T),(x^{w},x^{l},c)\in\mathcal{P}}[\log\frac{S(x _{t}^{w},c)^{\eta}}{S(x _{t}^{w},c)^{\eta}+S(x _{t}^{l},y)^{\eta}}+\alpha \cdot log\frac{\hat{D} _{s}(x _{t}^{w},c,s)}{\hat{D} _{s}(x _{t}^{l},c,s)}]$
> >
> >
> > |        | $t\in[501, 1000]$ | $t\in[1, 500]$ | #(Clean Image) |
> > |--------|-------------------|----------------|----------------|
> > | SLRM   | 66.35             | 66.59          | 67.08          |
> > | R-SLRM | 61.08             | 63.82          | 69.31          |
> >
> > However, when compared with our proposed formulation, this additive regularization leads to noticeably lower discrimination accuracy on noisy latents. This indicates that directly adding the score term does not provide sufficient noise-compatible preference signals and is less effective than our multiplicative score-enhancement design.  As an independent regularization, $\hat{D}_s$ fails to provide the necessary **denoising viability** to $S$. The model attempts to satisfy two separate and potentially conflicting objectives, causing the reward model to revert to its noise-sensitive nature when facing $x_t$.
> >
> > [**R-Q3：Strategy for Avoiding Rewared Hacking**]
> >
> > Thank you for the feedback. Although the random slection can be used to prevent the "reward hacking", our selection strategy is more effective for optmization. Specifically, during the dual-path sampling process, as described in Sec. 3.4 (Line 330–331), **selecting the best and worst samples at each step enables the algorithm to obtain pronounced quality differences**, which is essential for preference optimization. Replacing this mechanism with random selection removes this contrast, making SLRM’s evaluation much less informative.
> >
> > As for the combination setting, in our understanding, TAPO maintains a single sampling path. At each SDE sampling step, it still collects the best and worst noisy latents, but then randomly selects one latent into the next timestep. This degenerates TAPO into a variant similar to LPO, where the sampled candidates—especially when $P$ is small—can not obtain the pair with pronounced preference differences.
> >
> > In addition, we explore the influence of the pairwise combination with P= {2,4,6,8} in Table below. The results further confirm our intuition: although the combination setting avoids reward hacking, its performance remains inferior to our proposed method, and does not sample the strong preference contrast that TAPO relies on.
> >
> > |                           | HPSv3 | GenEval | FID   | LPIPS |
> > |---------------------------|-------|---------|-------|-------|
> > | TAPO (combination, P = 2) | 8.83  | 57.94   | 17.61 | 0.42  |
> > | TAPO (combination, P = 4) | 8.90  | 58.05   | 17.69 | 0.42  |
> > | TAPO (combination, P = 6) | 9.08  | 59.32   | 19.53 | 0.49  |
> > | TAPO (combination, P = 8) | 9.52  | 63.71   | 20.07 | 0.64  |

---

> ### Author Response · Authors · 2025-11-25
>
> Dear Reviewer,
>
> We hope this message finds you well. We sincerely appreciate the time and effort you have dedicated to reviewing our submission. We have now submitted our rebuttal and would like to kindly check whether our responses have adequately addressed your concerns.
>
> If you have any remaining questions or would benefit from further clarification, please feel free to let us know. We greatly value your feedback and are committed to improving the clarity and quality of our work.
>
> Thank you again for your thoughtful insights and constructive guidance.
>
> Best regards,

---

### Author Response · Authors · 2025-12-02

We sincerely thank the AC and all reviewers for their careful reading, constructive comments, and thoughtful feedback. We greatly appreciate the reviewers' **recognition of our paper's motivation, technical novelty, and empirical performance**, as well as the insightful questions that helped us further improve the clarity, analysis, and validation of our work.

We are glad to have successfully **addressed Reviewer yDbo's concerns** regarding SLRM generalization and qualitative gains, and are delighted that **they subsequently raised the score to 8**.  And We hope our rebuttal and new experiments similarly resolve the concerns of Reviewer PpWQ and Reviewer eNSr.


\
Below we summarize the reviewers' feedback and the comprehensive updates we have made to address the comments.
**1. Innovation & Motivation:**

- "The proposed reward model, SLRM, is innovative and efficient." (**PpWQ**)

- "The integration of the diffusion score function into reward estimation (SLRM) is original and theoretically grounded." (**yDbo**)

- "Identifies a genuine gap in current diffusion preference optimization (noise robustness and trajectory consistency) and addresses both with well-justified solutions." (**yDbo**)

**2. Effectiveness:**

- "Experiments show strong alignment performance on T2I and T2V tasks." (**PpWQ**)

- "The improvements over baselines are consistent across multiple preference and alignment metrics." (**eNSr**)

**3. Practicality:**

- "The proposed TAPO strategy provides a practical way to account for the entire sampling trajectory." (**PpWQ**)

- "TAPO's multi-timestep evaluation design is practical and effective." (**yDbo**)


\
We have addressed the primary concerns regarding additional baselines, computational cost, SLRM interpretability, and qualitative analysis through the following updates:

**1. Comprehensive Baseline Comparisons & Hybrid Strategies**
- **New Baselines**: We have added quantitative comparisons with **DenseReward (PpWQ)**, **SSPO, and Diffusion-RainbowPA (eNSr)**, demonstrating TAPO's superority.

- **Exploratory Analysis on Hybrid Reward Models**:  We explored combining SLRM with pixel-level rewards (HPSv3, PickScore). While combining with HPSv3 yields moderate gains, we analyzed the efficiency-effectiveness trade-off to address the concern about high-frequency details (**PpWQ**).

- **Investigation of Regularization Strategies**：We experimentally verified that our multiplicative score enhancement is superior to additive regularization provided by **PpWQ**, which failed to provide sufficient noise-compatible signals (**PpWQ**).

**2. Strengthened Interpretability & Empirical Validation of SLRM**

- **Generalization to Noisy Latents**: We clarified that the Score Enhancement mechanism ensures generalization by connecting preference scores to the generative prior. We validated this by showing TAPO maintains performance even when evaluating only partial timestep ranges (**yDbo**).

- **Interpretability**: We explained that "stable noise handling" is mathematically equivalent to "biasing toward easy-to-denoise samples," providing visualizations of reward distributions to show improved discrimination (**eNSr**).

**3. Efficiency, Scalability, and Quality Assurance**

- **Computational Cost**: We provided a detailed breakdown of training time and iteration costs, showing TAPO is comparable to other methods (like SPO and LPO) while delivering higher quality (**PpWQ**).

- **Trajectory Scaling**: We demonstrated that TAPO maintains efficiency even with longer diffusion trajectories ($T=20, 30, 40$) (**eNSr**) under the evaluation scaling linearly.

- **Reward Hacking & Quality**: We quantitatively identified the thresholds for reward hacking through ablations on candidate size ($n$) and pair size ($P$), and confirmed that our default settings successfully avoid reward hacking while maintaining image fidelity (**PpWQ**).

**4. Qualitative Clarifications**

- **Comparison with LPO**: We provided detailed analyses of Figure 5 and Figure 18, highlighting TAPO's specific advantages over LPO in terms of text alignment, detail consistency, and artifact reduction (**yDbo**).

\
In summary, we have taken all suggested baselines, ablation studies, and efficiency analyses into the consideration and made a revision. We believe these updates solidly validate the effectiveness of SLRM and TAPO.

---

### Meta-Review · Area_Chair_Tazs · 2026-01-08

**Summary:**

The paper initially received mixed ratings, including two borderline accepts and one borderline reject. All reviewers recognized the strong motivation, technical novelty, and solid empirical performance of the proposed preference optimization methods (SLRM and TAPO) for diffusion models. During the rebuttal, the authors provided a comprehensive and well-structured rebuttal, addressing the main concerns by adding additional baseline comparisons (PpWQ), conducting efficiency and quality assurance analyses (PpWQ, eNSr), providing interpretability and empirical validation of SLRM (yDbo, eNSr), and including qualitative analyses (yDbo). As a result, Reviewer yDbo upgraded his score from 6 to 8 before review reverting. Reviewer eNSr also confirmed via the follow-up email communication that all concerns were adequately addressed and upgraded his score from 4 to 6. Consequently, the paper now has unanimous positive ratings. Given that all major concerns have been resolved, we recommend acceptance of the paper.

**Reviewer Concerns:**

The authors provided a comprehensive rebuttal with more baseline comparisons with analysis results and interpretability clarification of the experimental results. Their rebuttal address Reviewer yDbo's concerns about SLRM generalization and qualitative gains and Reviewer eNSr's concerns about efficiency trade-off, interpretability of reward scores, more baseline comparisons with SSPO and RainbowPA. Both reviewers respectively upgrade their ratings from 6 to 8 before review reverting and from 4 to 6 through email communication for the final assessment. I think all the concerns are properly addressed.

**Reviewer Scores:**

The authors did provide a thorough and detailed rebuttal to address each concern raised by the reviewers. I believer the reviewers would upgrade or at least maintain the score if they had been able to participate fully in the discussion. From the author's rebuttal, Reviewer yDbo's agree to upgrade his score to 8 before reverting the score. With a follow-up email communication, Reviewer eNSr has confirmed that the authors have addressed his concern and agreed to upgrade his score to 6. The paper received all positive ratings after rebuttal.

---

### Decision · Program_Chairs · 2026-01-26

Accept (Poster)